# A comprehensive evaluation of the use of Lagrangian particle dispersion models for inverse modeling of greenhouse gas emissions

Martin Vojta[1], Andreas Plach[1,2], Rona L. Thompson[3], and Andreas Stohl[1]

[1]Department of Meteorology and Geophysics, University of Vienna, Vienna, Austria
[2]Physics Institute, Climate and Environmental Physics, University of Bern, Bern, Switzerland
[3]Norwegian Institute for Air Research NILU, Kjeller, Norway

**Correspondence:** Martin Vojta (martin.vojta@univie.ac.at)

**Abstract.** Using the example of sulfur hexafluoride ($SF_6$) we investigate the use of Lagrangian Particle Dispersion Models (LPDMs) for inverse modeling of greenhouse gas (GHG) emissions and explore the limitations of this approach. We put the main focus on the impacts of baseline methods and the LPDM backward simulation period on the *a posteriori* emissions determined by the inversion. We consider baseline methods that are based on a statistical selection of observations at individual measurement sites and a global distribution based (GDB) approach, where global mixing ratio fields are coupled to the LPDM back-trajectories at their termination points. We show that purely statistical baseline methods can cause large systematic errors, which lead to inversion results that are sensitive to the LPDM backward simulation period and can generate unrealistic global total *a posteriori* emissions. The GDB method produces *a posteriori* emissions that are far less sensitive to the backward simulation period and that show a better agreement with recognized global total emissions. Our results show that longer backward simulation periods, beyond the often used 5 to 10 days, reduce the mean squared error and increase the correlation between *a priori* modeled and observed mixing ratios. Also, the inversion becomes less sensitive to biases in the *a priori* emissions and the global mixing ratio fields for longer backward simulation periods. Further, longer periods might help to better constrain emissions in regions poorly covered by the global $SF_6$ monitoring network. We find that the inclusion of existing flask measurements in the inversion helps to further close these gaps and suggest that a few additional and well placed flask sampling sites would have great value for improving global *a posteriori* emission fields.

## 1 Introduction

Over the last few decades, the sharp increase of anthropogenic greenhouse gas (GHG) emissions has become a global concern, as it affects the Earth's climate with possible dangerous consequences for human health, infrastructure and ecosystems (IPCC, 2018). In order to prevent dangerous human interference with the climate system, the United Nations Framework Convention on Climate Change (UNFCCC) was established. As an important commitment to the convention, Annex-I countries (industrialized nations that are legally bound to reduce GHG emissions) are required to report their national emissions for regulated GHGs. These inventories are compiled by applying "bottom-up" methods, where statistical economic production or consumption data and source-specific emission factors are used to estimate national emissions. However, "bottom-up" estimates are suspected to suffer from significant uncertainties and there is a growing need for independent verification of these estimates (e.g. Rypdal

et al., 2005; Weiss et al., 2021). Independent verification can be provided by "top-down" methods, such as inverse modeling (e.g. Leip et al., 2017; Weiss and Prinn, 2011).

Inverse modeling requires the use of atmospheric transport models, either Eulerian models or Lagrangian Particle Dispersion Models (LPDMs). LPDMs are usually run backward in time. They release a large number of virtual particles from a given observation location and time, and trace them backward for a limited simulation period. The model output gives the sensitivity of the atmospheric mixing ratio to emissions during the backtracking time. In the inversion algorithm, the sensitivities for a large number of observations are used to optimize *a priori* emission estimates such that (with the obtained *a posteriori* emissions) the simulated mixing ratios better fit the atmospheric observations. Most studies only use continuous *in situ* observations for this purpose, however flask measurements with low sampling frequency can be included as well (e.g. Villani et al., 2010). For certain species, satellite measurements could also be used.

Previous studies argue that inversion methods have insufficient accuracy (e.g. Rypdal et al., 2005) and problems with reproducibility (Berchet et al., 2021). In order to enhance the credibility of inverse modeling, a better knowledge of the associated uncertainties is required (Brunner et al., 2017). An important source of uncertainty regarding LPDM-based inversion methods is the fact that they are often run backward in time only for a few days, e.g., 5 days (Keller et al., 2012; Vollmer et al., 2009; Zhao et al., 2009), 7 days (Koyama et al., 2011), 10 days (Schoenenberger et al., 2018; Simmonds et al., 2018; Thompson et al., 2017), or 20 days (Fang et al., 2014; Maione et al., 2014; Stohl et al., 2009). Koyama et al. (2011) and Stohl et al. (2009) are global inversion studies, while the other listed studies apply regional inversions. The choices of the used backward simulation period made by different authors seem arbitrary, and a systematic analysis of the impact of the backward simulation period is lacking.

The inversions can only account for the emissions that have occurred during the backward simulation period. By contrast, the emission contributions prior to the limited LPDM backward simulation period are not explicitly modeled but must still be accounted for in order to compare the model results with the observations. These contributions must be collected in a so-called baseline that is added to the modeled contributions. As errors in the baseline translate to errors in the *a posteriori* emissions, the baseline needs to be as accurate as possible. Many different methods have been suggested to determine this baseline.

Investigating halocarbons or fluorinated gases (F-gases) most studies use statistical methods to calculate the baseline by selecting low mixing ratio observations at individual stations (e.g. Ganesan et al., 2014; Prinn et al., 2000; Saito et al., 2010; Zeng et al., 2012). Such statistical methods have been operationally applied within observation networks, such as the Georgia Institute of Technology method (O'Doherty et al., 2001) used within the AGAGE community. The general idea is to statistically identify observations which are assumed to be unaffected by emissions within the LPDM simulation period. A widely used statistical method is the robust estimation of baseline signal (REBS) method, introduced by Ruckstuhl et al. (2012) which applies a robust local linear regression model. Statistical methods, however, always involve subjective data selection and treatment decisions, which can lead to problems. For instance, they will by definition wrongly classify measurements during longer lasting pollution episodes as baseline observations and therefore overestimate the baseline - a problem that is likely to occur frequently in polluted areas. It is also unclear to which degree these methods distinguish between lightly polluted air and measurement noise (Ryall et al., 2001). Furthermore, they fail to identify correct baseline mixing ratios when they are below

the lowest observations (Rigby et al., 2011), especially at polluted continental sites which virtually never receive air masses unaffected by emissions within the backward simulation period. In addition to the statistical selection some methods also use model information to improve the baseline. A method applied by the UK Met Office and commonly used within the AGAGE network (see e.g. Manning et al., 2021) identifies baseline measurements by analyzing the direction and height of air entering the regional inversion domain. A baseline method introduced by Stohl et al. (2009), further termed as "Stohl's method", uses model information to subtract prior simulated mixing ratios from pre-selected observations, in order to avoid an overestimation of the baseline. Nevertheless, this pre-selection is subjective and prior simulated mixing ratios depend on *a priori* emission estimates.

Apart from using observations at each individual station to maintain a baseline, Rödenbeck et al. (2009) suggested a general "nesting" scheme, where a regional transport model – either a Eulerian or Lagrangian model – is embedded into a global model providing information from outside the spatio-temporal inversion domain. Such a global distribution based (GDB) approach was used by many authors: Trusilova et al. (2010) and Monteil and Scholze (2021) used Rödenbecks approach to estimate $CO_2$ emissions. Similar Rigby et al. (2011) and Ganshin et al. (2012) developed approaches to nest a Lagrangian into a Eulerian model and tested it for $SF_6$ and $CO_2$, respectively. Estimating $CO_2$ baseline mole fractions for inverse modeling, Hu et al. (2019) applied two GDB approaches and a statistical method, where a subset of observations with minimal sensitivity was selected to correct a GDB baseline. Lunt et al. (2016) and Thompson and Stohl (2014) applied GDB approaches to model $CH_4$. While Thompson and Stohl (2014) coupled the LPDM back-trajectories with the global model at the end of the trajectories (which are terminated after a defined time), Lunt et al. (2016) used the exit location of the particles leaving the inversion domain for the coupling. The GDB method defines the baseline exactly in the way it is needed for the inversion and can account for meteorological variability (i.e., transport of air from regions with lower or higher mixing ratios, respectively), which may cause sudden changes in the baseline. The accuracy of the GDB method, however, depends on how well the global field of mixing ratios can be modeled.

The treatment of the baseline is critical when using LPDMs as a basis for atmospheric inversions. Still, it is unclear what influence the choice of a certain baseline approach has on inversion results. Previous studies indicated that different approaches lead to significant mismatches in simulated emissions (Thompson and Stohl, 2014; Henne et al., 2016). However, different methods were never compared systematically and tested for different model set-ups such as the length of the LPDM backward simulations.

Another problem of LPDM-based inversion studies is the general lack of consistency between regional emission estimates and the global emissions of a GHG. Given that the LPDMs are usually run backward in time only for a few days, the inversions constrain the emissions only in regions where observation stations exist (Rigby et al., 2011). This can lead to substantial deviations of the derived emissions from, often well known, global totals, a problem shared with regional inversion studies based on Eulerian models.

In this study we [i] investigate the effect of the backward simulation time period within the range of 0-50 days, [ii] analyze the impact of the baseline definition on inversion results, [iii] examine their consistency with known global total emissions, [iv] explore the influence of biases in the baseline and *a priori* emissions on inversion results for different backward simulation

periods, and [v] compare the value of different observation types (flask vs. continuous) for the inversion. We compare three different baseline methods - the REBS method, Stohl's method and the GDB method - and apply inverse modeling to the species sulfur hexafluoride ($SF_6$). $SF_6$ is the most potent GHG regulated under the Kyoto Protocol with a high global warming potential of approximately 23,500 over a 100-year time horizon (Myhre et al., 2013) and an estimated atmospheric lifetime of 3200 years (Ravishankara et al., 1993). $SF_6$ is a convenient choice for our studies because it has no negative sources (as,
e.g., $CO_2$), a very long lifetime in the atmosphere, well known global emissions, and there are relatively many measurements available. However, we expect our findings to also hold for other species and be informative for inverse modeling of GHGs with LPDMs in general.

## 2    Methods

### 2.1    Measurement data

The inversion (subsection 2.2) is performed by using continuous atmospheric observations of $SF_6$ dry-air mole fractions from 18 observation sites, distributed around the globe. Those measurements were provided by the Advanced Global Atmospheric Gases Experiment (AGAGE,  Prinn et al., 2018) network, the NOAA/ESRL halocarbons *in situ* program (Dutton et al., 2017) and a number of independent organisations whose data were partly included in the World Data Centre for Greenhouse Gases (WDCGG, 2018). Measurement sites are listed in Table 1, together with acronyms and other station specific information.

At AGAGE stations, $SF_6$ mixing ratios are measured using Medusa Gas Chromatography followed by Mass Spectrometry (GC/MS,  Miller et al., 2008). At the stations HAT and COI the $SF_6$ measurement system is based on cryogenic preconcentration and capillary GC/MS (Yokouchi et al., 2006). At all other stations, Gas Chromatography followed by Electron Capture Detection (GC-ECD) is used to measure $SF_6$ mole fractions. Observations were calibrated with four different $SF_6$ scales: SIO-2005, WMO SF6 X2006, WMO SF6 X2014 and NIES-2008. We converted all observations to the SIO-2005 calibration
scale, by dividing NIES-2008 calibrated data by the factor 1.013 (Saito, 2021) and WMO SF6 X2014 calibrated data by 1.002 (Guillevic et al., 2018). To convert mole fractions from WMO SF6 X2006 to WMO SF6 X2014, we used $y = ax^2 + bx + c$, where $y$ corresponds to $SF_6$ mole fractions on the X2014 scale, $x$ to mole fractions on the X2006 scale and the coefficients $a$, $b$, $c$ have the values of $2.6821 \cdot 10^{-3}$, $9.7748 \cdot 10^{-1}$, and $3.5831 \cdot 10^{-2}$ (NOAA ESRL, 2014), respectively.

We averaged all observation data over 3-hourly intervals. For stations at low altitudes, we selected afternoon values (12:00 to
16:00 local time), to only consider time periods with a well-mixed planetary boundary layer, when the smallest model errors can be expected. At mountain stations we instead selected observations during night times (00:00 to 04:00 local time) to avoid larger errors due to daytime small-scale up-slope winds in the complex topography around these sites, which are unresolved in the model. Additionally, we followed a method by Stohl et al. (2009) to identify observations that cannot be brought into agreement with modeled mixing ratios by the inversion, which we removed completely (in contrast to Stohl et al. (2009), who assigned
larger uncertainties to these observations). For this, we used the kurtosis of the *a posteriori* error frequency distribution and iteratively excluded observations causing the largest absolute errors until the kurtosis of the remaining error values fell below

**Table 1.** Sites of continuous surface measurements used in the inversion and in the re-analysis.

| Site ID | Station | Organisation | Calibration Scale | Latitude | Longitude | Altitude[a] | Frequency |
|---------|---------|--------------|-------------------|----------|-----------|-------------|-----------|
| CGO | Cape Grim, Tasmania | AGAGE | SIO-2005 | 40.7°S | 144.7°E | 94 | 2 hours |
| JFJ* | Jungfraujoch, Switzerland | AGAGE | SIO-2005 | 46.5°N | 8.0°E | 3580 | 2 hours |
| MHD | Mace Head, Ireland | AGAGE | SIO-2005 | 53.3°N | 9.9°W | 5 | 2 hours |
| RPB | Ragged Point, Barbados | AGAGE | SIO-2005 | 13.2°N | 59.4°W | 45 | 2 hours |
| SMO | Cape Matatula, American Samoa | AGAGE | SIO-2005 | 14.2°S | 170.6°W | 77 | 2 hours |
| THD | Trinidad Head, USA | AGAGE | SIO-2005 | 41.0°N | 124.1°W | 107 | 2 hours |
| ZEP | Zeppelin, Ny-Alesund, Norway | AGAGE | SIO-2005 | 78.9°N | 11.9°E | 474 | 2 hours |
| GSN | Gosan, South Korea | KNU/AGAGE | SIO-2005 | 33.3°N | 126.2°E | 89 | 2 hours |
| RGL | Ridge Hill, UK | UNIVBRIS | SIO-2005 | 52.0°N | 2.5°W | 204 | 30 min |
| ZSF* | Zugspitze-Schneefernerhaus, Germany | UBAG | WMO SF6 X2006 | 47.4°N | 11.0°E | 2671 | 1 hour |
| BRW | Barrow, Alaska, USA | NOAA | WMO SF6 X2014 | 71.3°N | 156,6°E | 11 | 1 hour |
| MLO* | Mauna Loa, USA | NOAA | WMO SF6 X2014 | 19.5°N | 155.6°W | 3397 | 1 hour |
| NWR* | Niwot Ridge, USA | NOAA | WMO SF6 X2014 | 40.0°N | 105.6°W | 3523 | 1 hour |
| SPO | South Pole, Antarctic | NOAA | WMO SF6 X2014 | 90.0°S | 24.8°W | 2841 | 1 hour |
| SUM | Summit, Greenland | NOAA | WMO SF6 X2014 | 72.6°N | 38.5°W | 3238 | 1 hour |
| IZO* | Izaña, Tenerife, Spain | AEMET | WMO SF6 X2014 | 28.3°N | 16.5°W | 2373 | 1 hour |
| COI | Cape Ochiishi, Japan | NIES | NIES-2008 | 43.2°N | 145.5°E | 49 | 1 hour |
| HAT | Hateruma, Japan | NIES | NIES-2008 | 24.1°N | 123.8°E | 47 | 1 hour |

[a] The altitude specifies the sampling height in meters above sea level. Stations considered as mountain sites are marked with an asterisk.

5 (close to a Gaussian distribution). This method removed 0.62% (63 data points) of the whole dataset, affecting 0 to 2.92% of the observations at individual measurement sites. In total, 10,142 observations were used in the inversion for the year 2012.

In order to generate global $SF_6$ mixing ratio fields required by the GDB method, we performed a two-year $SF_6$ re-analysis
(for more details see section 2.5), for which we used all the available 2011 and 2012 continuous measurements from the sites listed in Table 1. In addition, we included flask air samples from 44 surface observation stations (NOAA, Dlugokencky et al., 2020) and from 16 aircraft profiling stations (Sweeney et al., 2015; NOAA Carbon Cycle Group ObsPack Team, 2018). Surface flask measurements were available at intervals ranging from a few days up to months. Sampling flights were conducted irregularly with intervals between 2 and 5 weeks at individual sites. Aircraft measurements from individual flights provide vertical
$SF_6$ mixing ratio profiles up to 8.5 km above sea level, where air samples are usually taken within less than an hour. With one exception, all aircraft samples were collected over North America. Additional information about the flask measurements from surface sites and aircraft programs can be found in Table A1 and Table A2 (Appendix). All flask measurements were calibrated with the WMO SF6 X2014 calibration scale and we converted them to the SIO-2005 calibration scale. For the re-analysis, we used 175,557 in-situ, 3,423 surface flask, and 5,581 aircraft measurements amounting to 184,561 measurements in total in 2011
and 2012. Fig. 1 provides an overview of all observation sites considered in the inversion and the re-analysis. In one specific test case (see section 3.2) we also used the 2012 surface flask measurements in addition to the continuous measurements for the inversion.

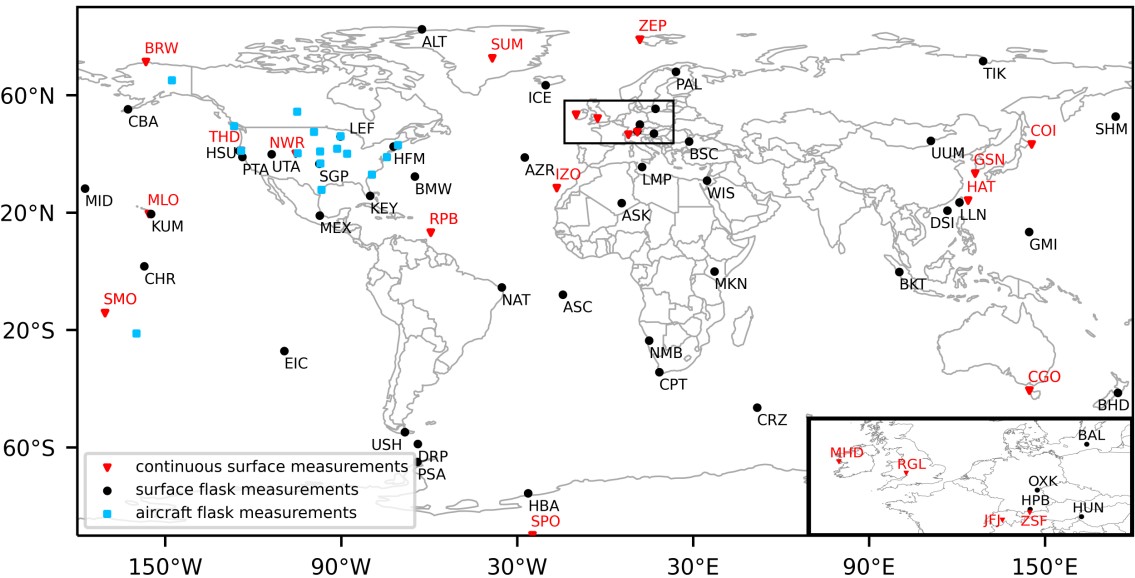

**Figure 1.** Map of sites with continuous surface measurements used for the inversion (red triangles) and flask measurements (surface: black dots, aircraft: blue squares) that were additionally used for the re-analysis of SF$_6$

## 2.2 Inversion method

In this study we use the Bayesian inversion framework FLEXINVERT+ described in detail by Thompson and Stohl (2014),

which was further developed since then, to make the code more modular and to include iterative solution methods. However, our results should be valid for all inversion methods based on LPDM calculations and we thus only include a brief description of FLEXINVERT+. It is based on a linear forward operator **H** that represents the atmospheric transport, so that the forward problem reads:

$$\mathbf{y} = \mathbf{H}\mathbf{x} + \varepsilon \tag{1}$$

where **y** is the vector of observed mixing ratios, **x** the emission state vector and $\varepsilon$ the sum of observation and model error. Since **H** is ill conditioned and has no unique inverse, *a priori* emission estimates can be added, in order to solve (1) for

**x**. The inversion method applies Bayes' theorem to calculate *a posteriori* emissions, which on the one hand minimize the difference between observed and modeled mixing ratios, and on the other hand stay close to the *a priori* emissions and inside of predefined uncertainty bounds. Assumed uncertainties are Gaussian distributed resulting in a minimization of the cost function (e.g. Tarantola, 2005)

$$\mathbf{J}(\mathbf{x}) = \frac{1}{2}(\mathbf{x} - \mathbf{x}_p)^T \mathbf{B}^{-1}(\mathbf{x} - \mathbf{x}_p) + \frac{1}{2}(\mathbf{H}\mathbf{x} - \mathbf{y})^T \mathbf{R}^{-1}(\mathbf{H}\mathbf{x} - \mathbf{y}), \tag{2}$$

where **B** is the *a priori* emission error covariance matrix, **R** the observation error covariance matrix, and $\mathbf{x}_p$ the vector of the *a priori* emissions. This study uses the following analytical solution to minimize $\mathbf{J}(\mathbf{x})$:

$$\mathbf{x} = \mathbf{x}_p + (\mathbf{H}^T \mathbf{R}^{-1} \mathbf{H} + \mathbf{B}^{-1})^{-1} \mathbf{H}^T \mathbf{R}^{-1}(\mathbf{y} - \mathbf{H}\mathbf{x}) \tag{3}$$

We use a spatial emission grid (Fig. A1) with 6219 grid cells of varying size ranging from $1° \times 1°$ to $16° \times 16°$. We define the grid by using model information to aggregate grid cells with low emission contributions, as further described by Thompson and Stohl (2014). For this, the emission sensitivity is taken from the LPDM 50-day backward simulation and the resulting inversion grid is used for all inversions. The output emission fields are saved at a spatial resolution of $1° \times 1°$. **x** is assumed to not vary with time.

SF$_6$ has no surface sinks and its surface fluxes can therefore only be larger or equal to zero. However, the inversion algorithm can produce negative *a posteriori* fluxes. To overcome this problem we follow Thompson et al. (2015) and apply an inequality constraint on the *a posteriori* emissions, using the truncated Gaussian approach by Thacker (2007). This approach, which applies inequality constraints as error-free observations, is described by the following equation:

$$\hat{\mathbf{x}} = \mathbf{x} + \mathbf{A}\mathbf{P}^T(\mathbf{P}\mathbf{A}\mathbf{P}^T)^{-1}(\mathbf{c} - \mathbf{P}\mathbf{x}), \tag{4}$$

where **P** is a matrix operator selecting the fluxes violating the inequality constraint, and **c** a vector of the inequality constraint (zero in our case). **x** and **A** represent the *a posteriori* emissions and error covariance matrix precalculated in the inversion, respectively.

In contrast to many other studies (e.g., Henne et al., 2016; Rigby et al., 2011; Stohl et al., 2009; Thompson and Stohl, 2014) we do not use the option to optimize the baseline mixing ratios in the inversion, except for sensitivity tests. In any case, it is desirable to obtain a baseline that is as accurate as possible prior to any optimization, which is a purely statistical correction that may falsely compensate for errors elsewhere (e.g., in the emissions). Waiving this option gives us further the opportunity to better analyze the differences between investigated baseline methods and to study their impacts on the *a posteriori* emissions more systematically. For the baseline optimization of the sensitivity tests, we use a temporal window of 28 days and a baseline uncertainty of 0.1 ppt. Increasing the uncertainty up to 0.2 ppt did not show any significant changes in the results. For general details on the baseline optimization see Thompson and Stohl (2014).

## 2.3 Atmospheric transport

**H** is the so-called Source-Receptor-Relationship (SRR) in the context of atmospheric transport. The SRR is an emission sensitivity that relates emission changes in a given grid cell to changes in modeled mixing ratios at a given receptor; for further

details, see Seibert and Frank (2004). The SRR value in a specific grid cell (units of $1\frac{sm^3}{kg}$) measures the simulated mixing ratio change at a receptor that a unit strength source ($1\frac{kg}{sm^3}$) in that grid cell would create (Stohl et al., 2009).

In this study, we use the LPDM FLEXPART 10.4 (Pisso et al., 2019; Stohl et al., 1998, 2005) to calculate the SRR. The model is run in backward mode as this is more efficient than forward calculations when the number of emission grid cells exceeds the number of observation sites. Available observations are averaged to three hourly means (see section 2.1). For each of these means 50,000 virtual particles are released continuously over the averaging period and followed backward in time. The SRR is calculated by determining the average time the particles spend in each grid cell of the $1° \times 1°$ output grid within the lowest 100 meters above the ground, assuming that all emissions occur at or near the ground. FLEXPART is driven by the hourly re-analysis dataset ERA5 (Hersbach et al., 2018) from the European Centre for Medium-Range Weather Forecasts (ECMWF) at a resolution of $0.5° \times 0.5°$ and with 137 vertical levels. Since $SF_6$ is an almost nonreactive gas, removal processes are neglected in the calculation of the SRR.

In this study, five different backward calculation periods are investigated: 1, 5, 10, 20 and 50 days. At the end of these periods, particles are terminated and the back trajectories end. Figure 2 shows the 2012 annual average emission sensitivities for the backward calculation period of 5 (Fig. 2a) and 50 (Fig. 2b) days, respectively. On the 5-day time scale large land areas in the Southern Hemisphere (Northern Australia, South America, Southern Africa) and also parts of the Northern Hemisphere (e.g. India, Iran) are sampled poorly or not at all. In these areas, emissions can therefore not be determined well by the inversion. High sensitivity can only be found at land regions with many receptors, such as Europe. On the 50-day time scale, the SRR has higher values compared to the 5-day backward calculation. Large parts of the Northern Hemisphere are sampled quite well and the emission sensitivities provide some information, even at areas that are far away from the observation stations. However, emission sensitivities are still low in the Tropics, especially over Africa, South America and Northern Australia. Figure 2c shows the increase in the annual averaged SRR due to the use of flask measurements in addition to continuous measurements in the case of 50-day simulations. One can see substantial increases in the vicinity of the measurement sites, that quickly decline with distance to the sites. Further SRR values increase in large parts of the Southern Hemisphere, however, the increases over southern continental areas are relatively low, as most flask measurements are not well located for inversion purposes.

## 2.4 The baseline definition

The transport model can only account for mixing ratio changes caused by emissions within the chosen backward calculation period. Consequently, a baseline representing the influence of all the emission contributions prior to this time period has to be defined.

### 2.4.1 The REBS method

The REBS method introduced by Ruckstuhl et al. (2012) is a statistical method using a robust local regression model to identify background observations from each individual observation station to estimate a baseline curve. In recent years it has been used in various studies to determine a baseline for atmospheric inversions of several GHG species (e.g. An et al., 2012; Brunner et al., 2017; Henne et al., 2016; Schoenenberger et al., 2018; Simmonds et al., 2016; Vollmer et al., 2016). The REBS method

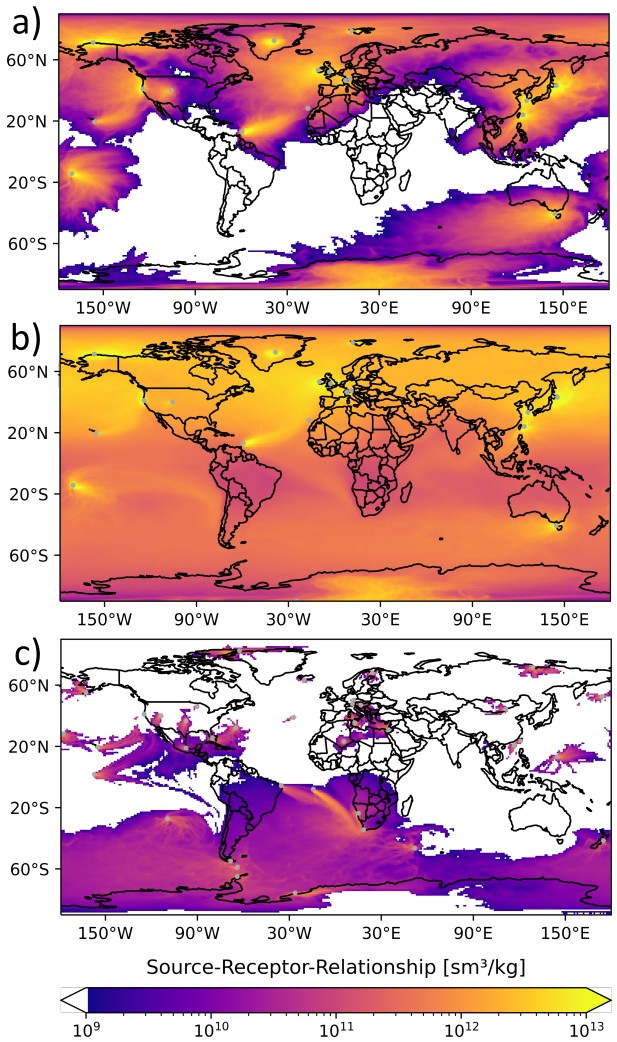

**Figure 2.** Source-Receptor-Relationship obtained from FLEXPART backward simulations, averaged over the year 2012. a) and b) show the SRR for all considered continuous measurement stations and for a simulation period of a) 5 and b) 50 days. c) shows the increase in the annual averaged SRR due to the use of flask measurements in addition to continuous measurements for the case of a 50-day backward simulation period.

defines observed mixing ratios $y(t_i)$ at each time step $t_i$ as the sum of a baseline signal $g(t_i)$, an enhancement due to polluted air masses $m(t_i)$ and the observational error $E_i$:

$$y(t_i) = g(t_i) + m(t_i) + E_i \tag{5}$$

The method assumes that most observations are baseline observations and therefore not influenced during pollution episodes ($m(t_i) = 0$). It also assumes that the baseline curve $g$ is smooth - so that it can be linearly approximated around any given

time. The method then applies a local linear regression model that fits the observation data, giving more weight to data points close to the considered time and iteratively excluding data points outside a certain range. An advantage of the REBS method is that it is simple to implement. The code is freely available and besides some parameters that need to be chosen, it only depends on the observation data. This simplicity, however, also means that the method is unable to take the length of the LPDM backward calculation into account. As we shall see, this leads to systematic biases in the inversion results that depend on the length of the backward calculation. The method also assumes a smoothly varying baseline which limits its ability to account for meteorological variability. Another disadvantage is the dependence on certain parameter settings. The settings used in this study are provided in Table A3. Finally, the method can only be used at sites with frequent observations, not for flask measurement sites, or moving measurement platforms.

### 2.4.2 Stohl's method

The method introduced by Stohl et al. (2009) is primarily based on the selection of observed mixing ratios at individual observation stations, but also uses the simulated SRR values and *a priori* emissions to determine the baseline. In the last few years it was used in several inversion studies (e.g. Brunner et al., 2017; Fang et al., 2014, 2015, 2019; Stohl et al., 2010; Thompson and Stohl, 2014). We apply the method and select the lowest 25 percent of observations from individual stations in a moving time window of 30 days to only consider observations which are weakly influenced by emissions within the backward calculation period. Prior simulated mixing ratio enhancements are subtracted from the selected observations to eliminate the emission contributions from within the time interval of the LPDM simulation. In order to avoid an overestimation of their contribution, only the lower half of the prior simulated values and the corresponding observed data points are selected. In every time window resulting mixing ratios are averaged and finally linearly interpolated to the timestamp of the observations. By subtracting prior simulated mixing ratios the method takes the length of the LPDM backward calculation into account and aims to avoid an overestimation of the baseline. However, simulated mixing ratios are calculated using *a priori* emission estimates, making the method dependent on *a priori* information. Further, the subjective choice of the time window, as well as the equally subjective selection of the lowest quartile of observations and the lower half of prior simulated mixing ratios are problematic. As the REBS method, Stohl's method assumes a smooth baseline curve, and thus it cannot account for sudden changes in the baseline due to meteorological variability. Also, the method can only be used at sites with frequent observations.

### 2.4.3 The GDB method

The idea of the GDB approach (Thompson and Stohl, 2014) is to determine the baseline directly from a 3D global field of mixing ratios, e.g., from a re-analysis of the atmospheric chemical composition. The end points of the back-trajectories that are used by the LPDM to calculate the SRR are utilized to determine the sensitivity at the receptor to mixing ratios at the points in space and time where particles terminate (see Fig. 3 for a simplified illustration). This sensitivity (termed as "termination sensitivity", thereafter) in a particular grid cell is calculated in the LPDM by dividing the number of particles terminating in that cell by the total number released at the receptor, while also including a transmission function to account for loss processes (not relevant for $SF_6$) during the backward simulation period. The termination sensitivity fields are saved in a 3D $1° \times 1°$ output

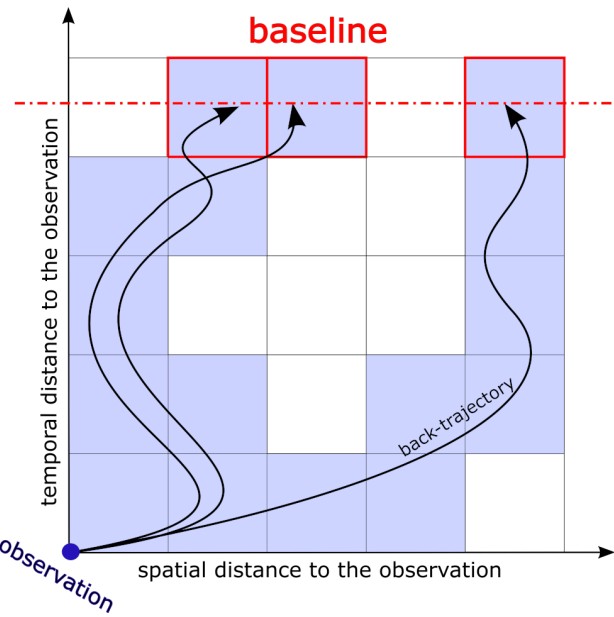

**Figure 3.** Simplified illustration of the global-distribution based (GDB) method for baseline determination, where the backward simulation is represented by three back trajectories released at the time and space of a particular observation. The spatio-temporal grid is simplified to two dimensions with a vertical time and a horizontal space axis. Grid cells that contribute to the modeled mixing ratio through emissions are shaded blue; termination grid cells where termination sensitivity is stored are marked with red rectangles; the termination point is illustrated by a red dashed horizontal line.

grid with 16 vertical layers with interface heights at 0.1, 0.5, 1, 2, 3, 4, 5, 7, 9, 12, 15, 20, 25, 30, 40, and 50 km above ground level. For global inversions, baseline mixing ratios are then calculated by multiplying the termination sensitivity with the mixing ratios of the 3D global field and integrating the product over all grid cells. The GDB method can also be used for regional inversions (not done in this study). In this case, the emission contributions from outside the regional domain need to be added to the baseline (Thompson and Stohl, 2014) but otherwise the inversion procedure is identical as described here.

The GDB method is independent of subjective data selection and choice of parameter settings. In contrast to the REBS and Stohl's method it does not assume a smooth baseline and has the potential to fully account for meteorological variability. As illustrated, it excludes emission contributions from within the backward simulation period and therefore provides a baseline that is fully consistent with the length of the backward simulation. Furthermore, contrary to the other two methods, it can also be used at measurement sites with infrequent observations or moving observation platforms. Its accuracy, however, is

dependent on the ability to minimize errors, and especially biases of the global 3D mixing ratio fields. We target this challenge by using the FLEXible PARTicle dispersion chemical transport model (FLEXPART CTM, Henne et al., 2018) to perform a re-analysis of $SF_6$ as described in the next section.

## 2.5 Re-analysis of SF$_6$ using FLEXPART CTM

In this study the LPDM FLEXPART 8-CTM-1.1 is used to perform a re-analysis of SF$_6$ for the year 2012. It was developed by Henne et al. (2018) and is based on FLEXPART 8.0. Groot Zwaaftink et al. (2018) provide a detailed description of FLEXPART CTM and evaluate this model for the example of CH$_4$. FLEXPART CTM is run in a domain filling mode where 12 million particles are randomly distributed over the globe, proportional to the air density. In addition to an air tracer, particles also carry the chemical species SF$_6$. The initialisation is based on a latitudinal SF$_6$ profile based on surface observations. We run the simulation from 2011 to 2012, using 2011 as a spin up period. Particles are followed forward in time, and whenever a particle resides below the diagnosed boundary layer height its mass is increased due to surface SF$_6$ emissions. The model is driven with the ECMWF ERA5 dataset and with emission fields calculated as described in section 2.6. Mixing ratio fields are saved daily on a $3° \times 2°$ output grid and coupled to the backward simulations.

FLEXPART CTM uses a nudging routine to keep simulated SF$_6$ fields close to the observations of SF$_6$. With this simple data assimilation method, modeled fields of mixing ratios are relaxed towards observations within so-called nudging kernels around observation sites. For all surface observation stations in the Southern Hemisphere we assign relatively large uniform kernel sizes, since the model tends to overestimate SF$_6$ mixing ratios in the Southern Hemisphere and there are only few measurement stations to correct this bias. For the surface observation sites in the Northern Hemisphere, we assigned smaller kernel sizes to measurement stations with a large observation variability to conserve SF$_6$ spatial variability, especially over the continents (see Groot Zwaaftink et al., 2018). For the aircraft measurements we pre-define vertical levels at 0.05, 0.15, 0.3, 0.5, 0.75, 0.1, 1.5, 2, 2.5, 3, 3.5, 4, 5, 6, 7, 8 and 9 km above ground level, co-locate the individual measurements to the closest vertical level and chose kernel sizes that increase with altitude. Specific kernel settings are detailed in Table A4.

## 2.6 *A priori* emissions

An *a priori* estimate of the spatial distribution of SF$_6$ emissions for the year 2012 is determined by collecting information on the emissions from individual countries. We use country emissions reported to the United Nations Framework Convention on Climate Change (UNFCCC, 2021) and for East Asian countries emissions estimated by Fang et al. (2014). The sum of these individual country emissions is subtracted from the total global SF$_6$ emissions determined by Simmonds et al. (2020) and the remaining emissions are distributed to all other countries proportional to their electric power consumption (World Bank, 2021). Finally, total country emissions are disaggregated according to the gridded population density (CIESIN, 2018) within each country's borders. Note at this point that the *a priori* emissions as constructed agree with recognized global emissions, which should be kept in mind when the global total is used as a reference value in the discussion. The *a priori* emission uncertainty is estimated to be 50% in each grid cell with a minimal value of $1 \cdot 10^{-13} \frac{kg}{m^2 h}$. Spatial correlation between uncertainties are considered by using an exponential decay model with a scale length of 250 km.

## 3   Results

### 3.1   Baselines and length of backward simulation

The three investigated baseline methods are discussed for the example of two measurement sites, Gosan and Ragged Point, and for five backward simulation time periods. The Gosan observation station is located on the south-western tip of the Korean Island Jeju, monitors the outflows from the Asian continent, and is respresentative of stations which frequently measure pollution events. The Ragged Point observation station is situated on the eastern edge of Barbados with direct exposure to the Atlantic Ocean. Ragged Point is primarily influenced by easterly winds providing "clean" background air masses, uninfluenced by local emissions, and is therefore representative of background stations. Both, Gosan and Ragged Point periodically intercept

air from the southern hemisphere and therefore have a rather complex baseline.

Baseline mixing ratios are plotted together with respective observations and *a priori* mixing ratios for different LPDM backward simulation periods ranging from 1 to 50 days (Fig. 4-7). *A priori* mixing ratios are calculated as the sum of the baseline and the contribution originating from *a priori* emissions during the period of the backward simulation (termed "direct

emissions contributions" thereafter). Ideally, the choice of the backward simulation period should have no systematic effect on the calculated *a priori* mixing ratios. By increasing the backward simulation time, and therefore enlarging the temporal domain, additional emission contributions are included in the optimization. Per definition, these contributions are not part of the baseline and should ideally be removed from it. As a result, the baseline should become lower and smoother when the simulation period is increased. We investigate the agreement between modeled and observed mixing ratios for the three

methods with time series plots (Fig. 4-7), as well as statistical parameters (bias, mean squared error (MSE), and coefficient of determination ($r^2$)) summarized in Table 2.

Figure 4 shows the smooth baselines calculated with the REBS and Stohl's method at the measurement station Gosan. In the case of 1-day backward simulations (Fig. 4a/d) both methods show a poor agreement between modeled and observed mixing ratios, as neither the smooth baselines nor the small direct emission contributions can reproduce the observed mixing ratios

during pollution episodes. This agreement becomes much better with longer backward simulation periods (Fig. 4b/e). The REBS baseline stays completely unchanged for different backward simulation periods. Therefore, *a priori* mixing ratios grow with increasing simulation periods (Fig. 4b/c), as more direct emissions contribute to the calculated total mixing ratio. For Gosan, the bias is negative for the 1-day simulation period but becomes increasingly positive for longer simulation periods (Table 2). This systematically increasing bias is inherent to all purely observation based baseline methods and cannot be

corrected without adding model information. In contrast, Stohl's baseline level decreases with longer backward simulation periods as higher direct emission contributions are subtracted from the pre-selected observations. Consequently, the bias of the *a priori* mixing ratios changes less between 10 and 50 days of backward simulation (Fig. 4e/f). This is confirmed by statistical parameters in Table 2, showing also only little change between 10 and 50 days.

At Ragged Point (Fig. 5) the *a priori* mixing ratios determined by the REBS method fit the observation data very well

for short backward simulation periods, where baseline and *a priori* mixing ratios overlap because of small direct emission contributions (Fig. 5a/b). This is expected, since the method determines the baseline by fitting the observation data, while

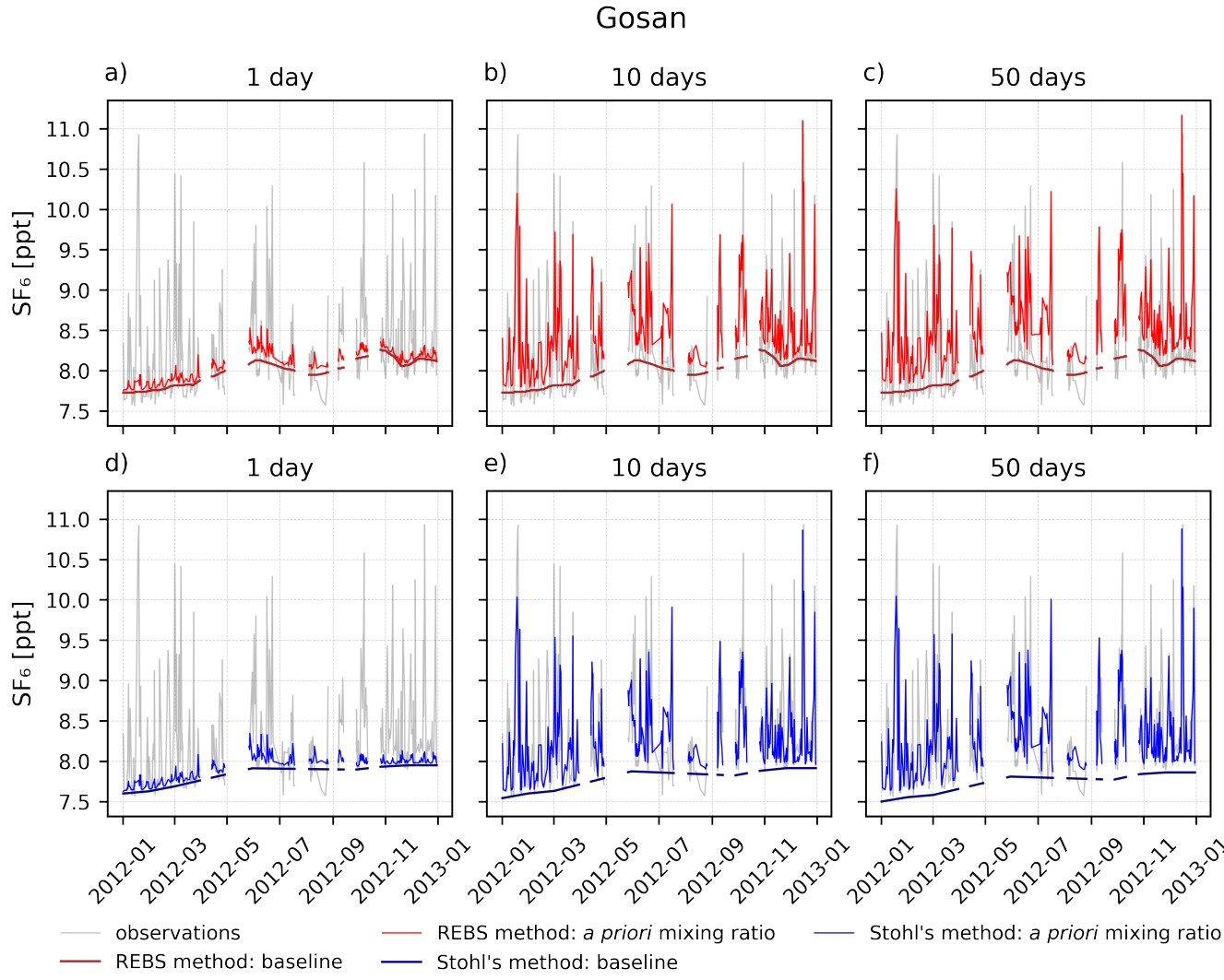

**Figure 4.** Baseline and *a priori* SF$_6$ mixing ratios calculated with the REBS (upper panels) and Stohl's method (lower panels) at the Gosan observation station, compared to SF$_6$ observations. Model results are shown for backward simulations of 1 day (panels a and d), 10 days (panels b and e) and 50 days (panels c and f).

**Table 2.** Bias, mean squared error (MSE) and coefficient of determination ($r^2$) of *a priori* SF$_6$ mixing ratios determined by the three investigated baseline methods with respect to observed mixing ratios. Statistical parameters are shown for three different backward calculation periods (1, 10, and 50 days) at the stations Gosan and Ragged Point. Also reported are the bias, MSE and $r^2$ calculated separately for all stations listed in Table 1, and then averaged.

| Parameter | method | Gosan 1d | Gosan 10d | Gosan 50d | Ragged Point 1d | Ragged Point 10d | Ragged Point 50d | all stations 1d | all stations 10d | all stations 50d |
|---|---|---|---|---|---|---|---|---|---|---|
| | REBS | - 0.225 | 0.190 | 0.267 | 0.006 | 0.007 | 0.054 | - 0.028 | 0.012 | 0.061 |
| Bias [ppt] | Stohl | - 0.384 | - 0.016 | 0.008 | - 0.067 | - 0.068 | - 0.065 | - 0.103 | - 0.064 | - 0.051 |
| | GDB | - 0.090 | -0.002 | -0.006 | 0.023 | 0.044 | 0.033 | 0.022 | 0.016 | 0.007 |
| | REBS | 0.420 | 0.250 | 0.281 | 0.004 | 0.004 | 0.006 | 0.034 | 0.023 | 0.028 |
| MSE [$ppt^2$] | Stohl | 0.525 | 0.216 | 0.210 | 0.009 | 0.009 | 0.009 | 0.050 | 0.026 | 0.024 |
| | GDB | 0.303 | 0.206 | 0.205 | 0.005 | 0.005 | 0.004 | 0.034 | 0.022 | 0.021 |
| | REBS | 0.085 | 0.482 | 0.495 | 0.671 | 0.670 | 0.712 | 0.584 | 0.647 | 0.651 |
| $r^2$ | Stohl | 0.068 | 0.474 | 0.490 | 0.649 | 0.629 | 0.623 | 0.548 | 0.616 | 0.623 |
| | GDB | 0.272 | 0.499 | 0.501 | 0.631 | 0.718 | 0.746 | 0.423 | 0.589 | 0.634 |

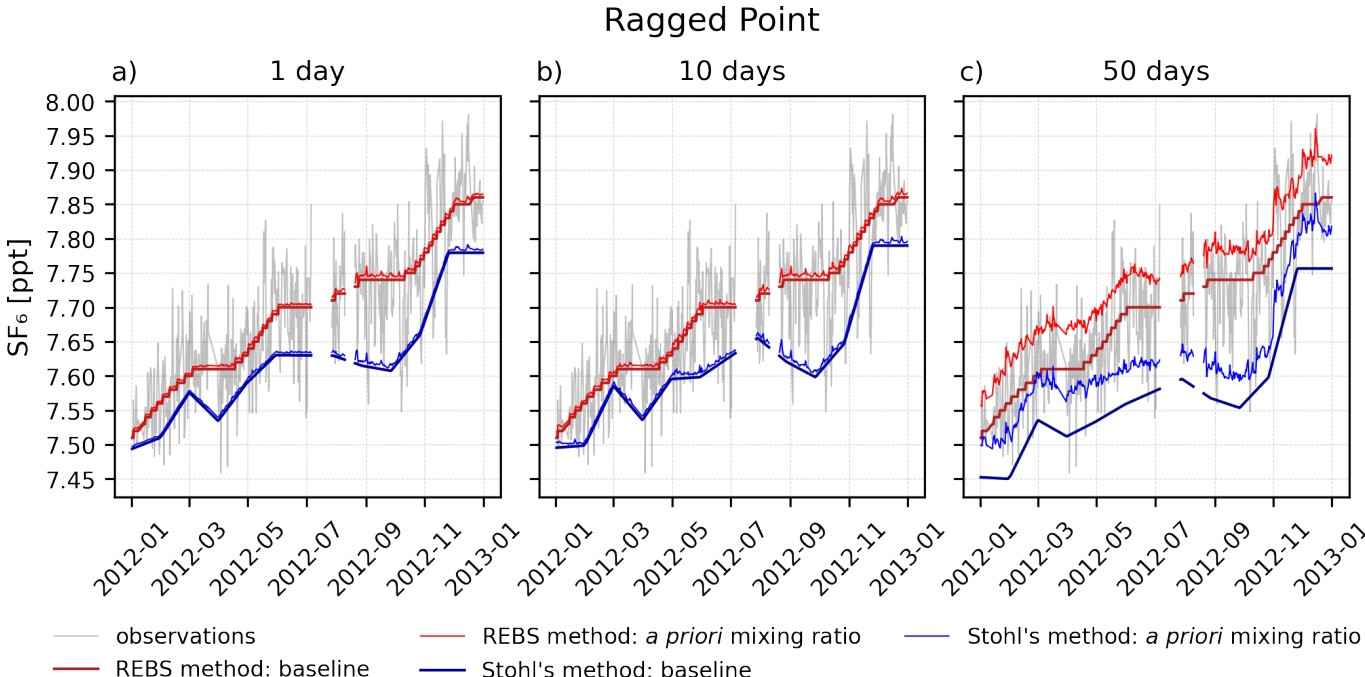

**Figure 5.** Baseline and *a priori* SF$_6$ mixing ratios determined by the REBS and Stohl's method at the Ragged Point observation station for backward simulation times of 1 day (panel a), 10 days (b) and 50 days (c).

iteratively excluding outliers. Since regional pollution events captured at Ragged Point tend to be very small, no significant measurement peaks need to be excluded. Therefore, the REBS baseline fits well through the measurement data, resulting in a good statistical model-observation agreement (Table 2). However, the smooth baseline is unable to reproduce the observed variability. In the case of a simulation period of 50 days (Fig. 5c), more direct emission contributions give higher *a priori* mixing ratios, overestimating the measurements and causing a large bias. In contrast, due to its 25th percentile pre-selection of observations, Stohl's method shifts the baseline curve towards the lowest observations. In the case of Ragged Point, these lowest observations come from southern hemispheric air masses. Hence, Stohl's baseline is more representative for southern hemispheric conditions, which do not necessarily dominate at that site. Consequently, *a priori* mixing ratios underestimate the observations for low direct emission contributions (Fig. 5a/b). The resulting bias is almost unaffected by the different backward simulation periods (Table 2, Fig. 5c), showing the method's ability to compensate for increasing direct emission contributions. However, the rather ad hoc 25th percentile pre-selection of data for the baseline is obviously not justified for a background station with few pollution episodes and southern hemispheric air interceptions, leading to a systematic underestimation of modeled *a priori* mixing ratios, irrespective of the length of the backward simulation.

The GDB method is illustrated for all tested backward simulation periods, including a case without any backward simulation (0 days). In this extreme case the baseline is obtained directly from the value of the global mixing ratio field simulated with FLEXPART CTM in the spatio-temporal grid cell of the respective observation. At Gosan, FLEXPART CTM reproduces observed mixing ratios well, even capturing a few pollution events (Fig. 6a). This good agreement is however expected, since these observations were used for the nudging in the FLEXPART CTM model. In the 1-day backward simulation case (Fig. 6b), the method computes a highly variable baseline partly representing the observed variability. This results in a much better agreement between *a priori* and observed mixing ratios than using the REBS or Stohl's method (Table 2). The GDB baseline becomes smoother and lower with increasing backward simulation time. The loss of variability arises from the fact that the GDB method calculates the baseline from a weighted average of grid cell mixing ratios at the trajectory termination points. The longer particles are followed backward in time, the more widely dispersed over large geographical regions termination points become, thus resulting in a smoother baseline. The lowering of the GDB baseline is compensated by the increase of the direct emission contributions (see Section 2.4.3 and Fig. 3), ensuring a seamless transition between forward (Flexpart CTM) and backward simulations. As a result *a priori* mixing ratios in Fig. 6 show no large systematic changes with increasing simulation period between 5 and 50 days.

Figure 6 also demonstrates the advantage of the Lagrangian backward simulation. As FLEXPART CTM is limited in resolution and particle number, it can only reproduce a few pollution events at Gosan, underestimates the highest and overestimates the lowest measured $SF_6$ mixing ratios, as demonstrated in the 0-day case (Fig. 6a). The backward simulation is initiated at the exact location of the measurement point and provides much higher resolution (Fig. 6b-f). If the backward calculation period is long enough that back trajectories reach important emission regions, mixing ratio spikes similar to the observed ones can be simulated. At the same time, the lowered baseline for intrusions of southern air masses during the Asian summer monsoon also allows capturing the lowest observed values. Table 2 shows exclusively improving correlation between modeled and observed values with increasing backward simulation periods.

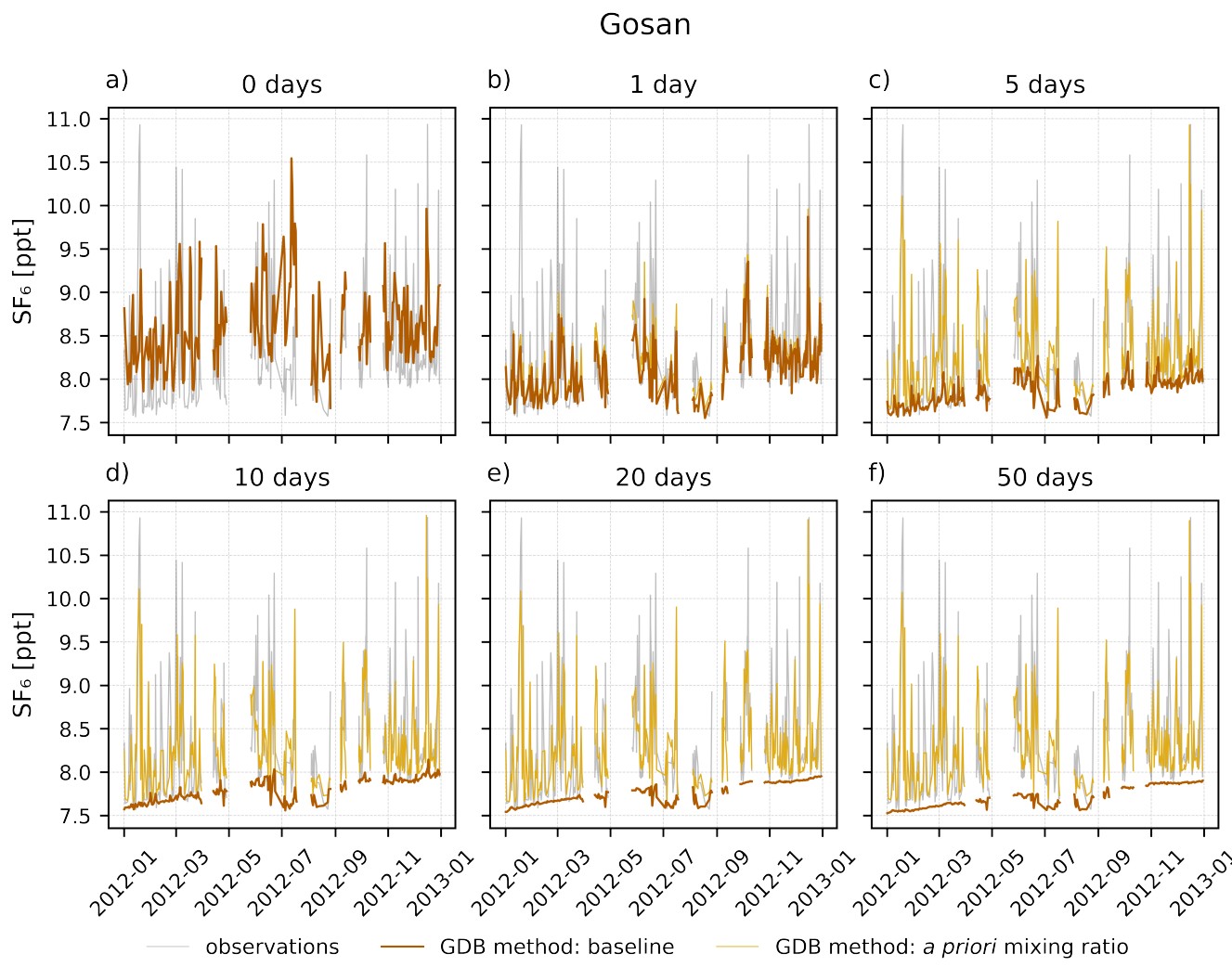

**Figure 6.** Baseline and *a priori* SF$_6$ mixing ratios calculated with the GDB method at the Gosan observation station for backward simulation times of 0 days (panel a), 1 day (b), 5 days (c), 10 days (d), 20 days (e) and 50 days (f).

# Ragged Point

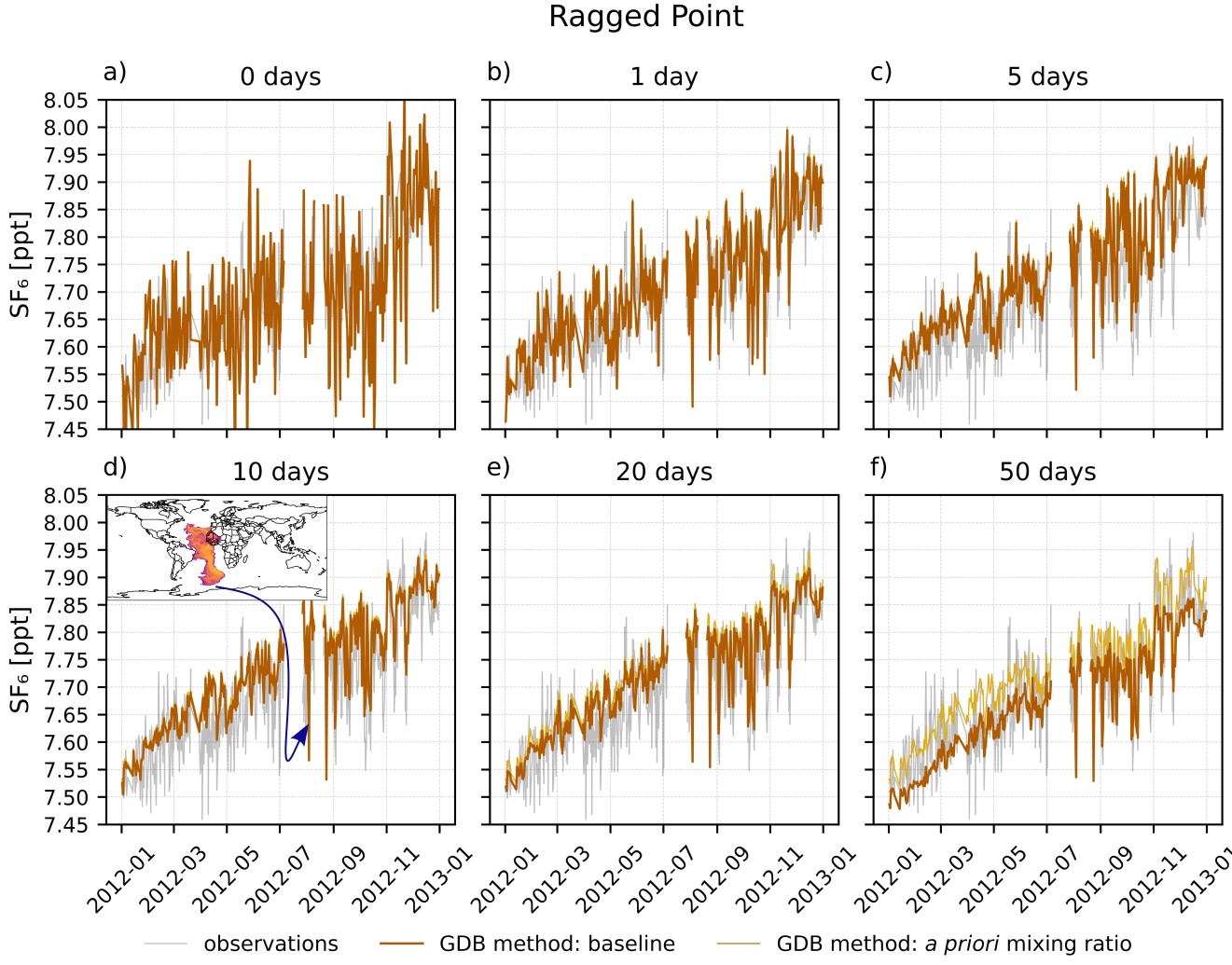

**Figure 7.** Baseline and *a priori* SF$_6$ mixing ratios calculated with the GDB method at the Ragged Point measurement station for backward simulation periods of 0 days (panel a), 1 day (b), 5 days (c), 10 days (d), 20 days (e) and 50 days (f). The inset in panel d shows the termination sensitivity averaged over all heights for the time of the marked observation low point, illustrating the methods ability to account for baseline changes, due to episodic transport from the Southern Hemisphere.

Figure 7 illustrates the GDB method at the Ragged Point station. FLEXPART CTM (Fig. 7a) reproduces the measured mixing ratios well. However, it generates more variability than observed at this station. This is partly due to the limited number of particles in the domain-filling simulation, which introduces noise into the model results. This is averaged out by
the GDB method with increasing backward simulation time, as the baseline becomes a weighted average over many grid cells. Nevertheless, the baseline maintains variability for all tested simulation periods, fitting the observed signal well (Fig. 7b-e). It is noteworthy that at Ragged Point a substantial part of the observed $SF_6$ variability seems to be caused by transport from different latitudes/regions without direct emission contributions, exemplified by the quite variable baseline even for the 50-day backward simulation. In contrast, the direct emissions accumulated over the 50 days of the backward simulation are producing
an almost constant enhancement over the baseline. This is very different from a station like Gosan that is strongly influenced by pollution episodes.

Notice also that for backward simulation times of 10 days and longer, the combined FLEXPART CTM/backward model is able to reproduce short episodes of very low observed mixing ratios at Ragged Point that are caused by episodic transport from the Southern Hemisphere (see also inset in Fig. 7d). Neither the REBS nor Stohl's method could correctly reproduce these
negative $SF_6$ excursions.

Additional figures illustrating the three baseline methods at all investigated measurement sites, can be found in the supplementary material. Despite of all advantages of the GDB method, it doesn't work well if the modeled global mixing ratio fields are biased. At Mace Head and Zeppelin (see supplementary Fig. S17 and S33), FLEXPART CTM overestimates the measurements, and thus the GDB method gives a baseline that partly exceeds the observations. Possible error sources include
deficiencies in the emission assumptions driving the model, that are impossible to be compensated through nudging with the few available observations. It is also unclear whether the FLEXPART CTM nudging routine was able to properly correct mixing ratios at higher altitudes, as aircraft measurements were available only over North America (with one exception). On the other hand, statistical baseline methods might work better at observation stations, where the baseline termination is less complex. At Mace Head (Fig. S18) for instance, both REBS and Stohl's method lead to a very high correlation between modeled
and observed mixing ratios for the case of a 50-day backward simulation ($r^2 = 0.87$). Nevertheless, for the REBS method, the discussed growing negative bias with longer simulation periods can be observed.

Statistical parameters (bias, MSE, and $r^2$) were separately calculated for every observation station and respective averages over all stations are shown in Table 2. One should keep in mind that the REBS and Stohl's method are directly based on the observations themselves and thus the dependency between observed and modeled *a priori* mixing ratios is likely higher
than in the case of the GDB method, where observations are rather used to improve the mixing ratio fields. Therefore, it is remarkable that overall the GDB method obtains smaller bias and MSE values than the other two methods. The REBS method shows the highest $r^2$ values. The main reason for this good correlation is, that the method captures the trend in the time series very well, which represents a considerable fraction of the total variability in the data. The GDB baseline may contain a fair fraction of noise, in contrast to the smooth baselines of the other two methods. This will lead to lower correlation. However, it
is noteworthy that for the GDB method, the $r^2$ value improves systematically with growing backward simulation time and for 50 days even exceeds the value derived by Stohl's method. By extending the backward calculation period from 10 to 50 days

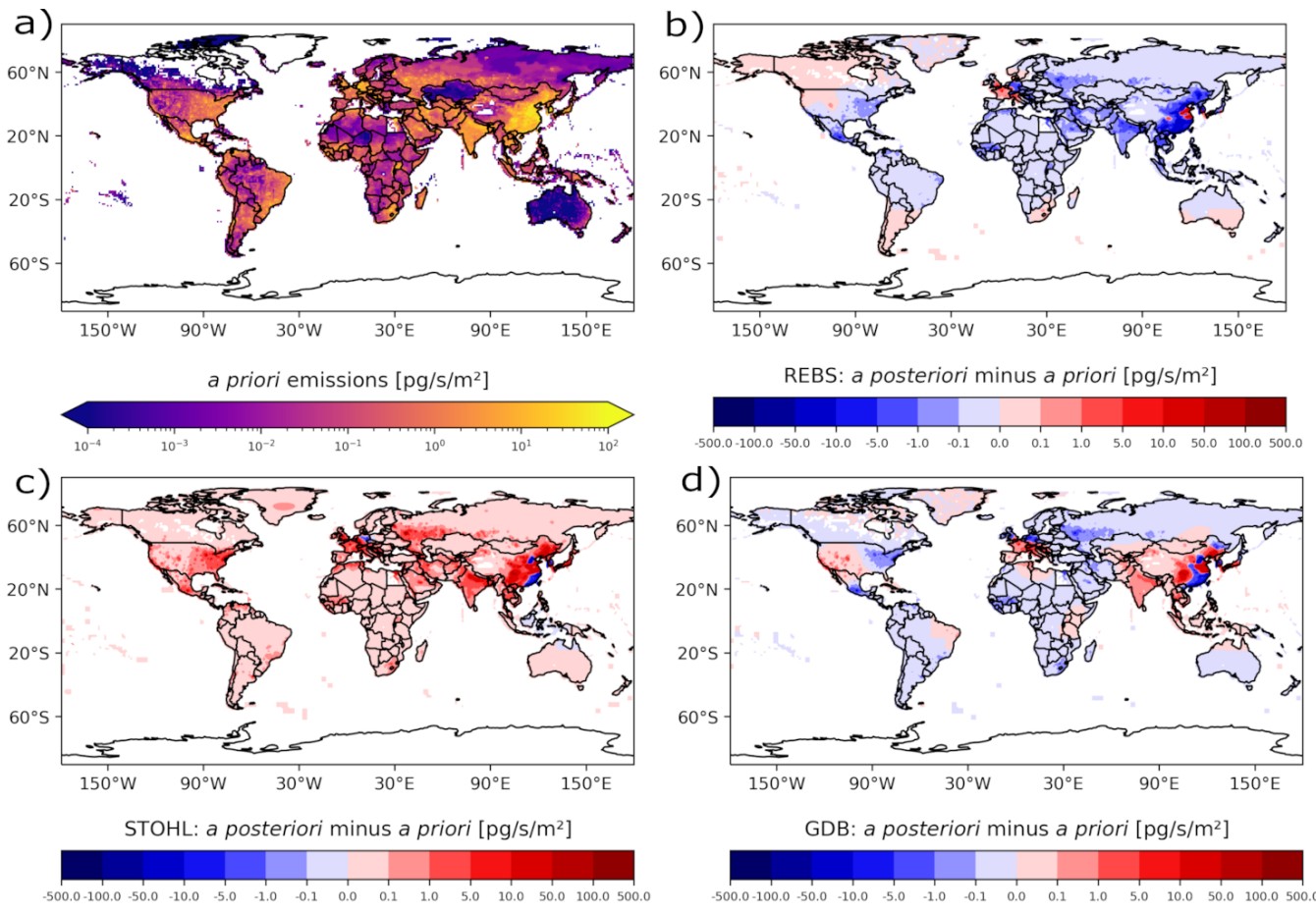

**Figure 8.** *A priori* SF$_6$ emissions (a) and SF$_6$ emission increments given by the inversion when using the REBS method (b), Stohl's method (c), and the GDB method (d) based on 20-day LPDM backward simulations.

the GDB $r^2$ value increases by 0.045, meaning that an extra 4.5% of the observed variability can be explained by the model. Notice also the improvement in bias and MSE, which can be observed for the GDB and Stohl's method, when extending the simulation period from 10 to 50 days. The REBS method does not show these improvements due to its systematical increase of bias with backward simulation time.

## 3.2 Inversion Results

Figure 8 illustrates (a) the global distribution of the SF$_6$ *a priori* emissions 2012, as well as (b-d) the emission increments (i.e., *a posteriori* minus *a priori* emissions) for the three investigated baseline methods using SRRs from 20-day backward calculations. *A priori* emissions are allocated to regions proportional to electricity use and population density. This implies large *a priori* emissions in South and East Asia, including China which is estimated to be the biggest contributor to global

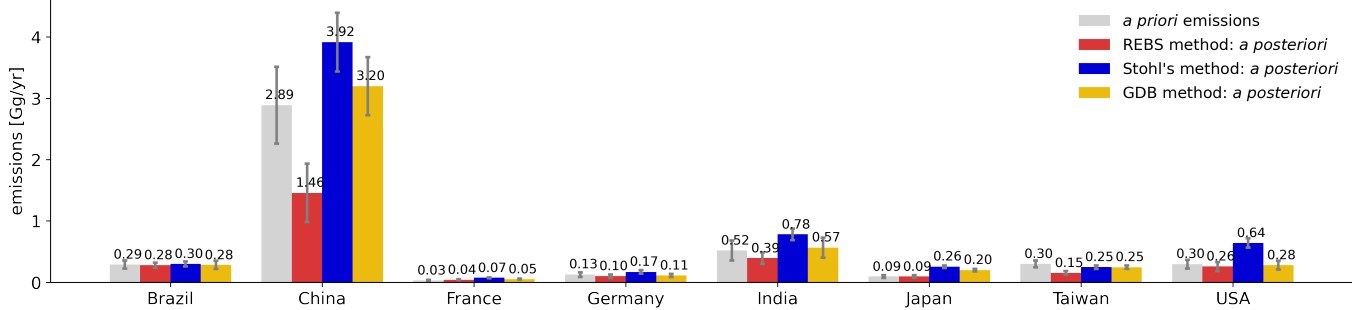

**Figure 9.** National SF$_6$ emissions for selected countries, based on 20-day LPDM backward calculations with different choices of the baseline method. Uncertainties represent a 1$\sigma$ range.

SF$_6$ emissions. In general, much higher *a priori* emissions are allocated to the Northern than to the Southern Hemisphere. We should also note that the emission optimization of the inversion focuses on regions with large *a priori* emissions, where also assumed uncertainties are bigger (see Section 2.6), assigning a larger degree of freedom to the algorithm.

The inversion increments in Fig. 8b-d show three very contrasting pictures, illustrating the huge impact of the choice of the
baseline method on the inversion results. Using different baseline approaches completely changes the results of the inversions. When using the REBS method (Fig. 8b), the inversion produces negative emission increments in almost all areas of the globe. As the real emissions are unknown, this is not necessarily an unrealistic result. However, when considering these mostly negative increments together with the discussed positive bias for REBS baselines in Table 2 (especially for longer backward simulation periods), there is reason to assume that the REBS method overestimates baselines and consequently underestimates
the *a posteriori* emissions overall. In contrast, the inversion algorithm produces positive increments almost everywhere around the globe when applying Stohl's method (Fig. 8c). Again, considering this together with the discussed negative biases in Table 2, this might indicate an underestimation of the baselines and an overestimation of the *a posteriori* emissions overall. In case of the GDB method (Fig. 8d) negative and positive increments are more balanced. Overall the patterns are more similar to the ones of the REBS method, except in East Asia, where they rather resemble the patterns of Stohl's method. Large positive
increments can be seen in East Asian regions and parts of Europe, whereas the inversion tends to produce slightly negative increments in the Southern Hemisphere.

**National emissions**

As the verification of emission reports to UNFCCC takes place on a national scale, the impact of baseline methods on national emissions is of great interest (Fig. 9). In countries with very low emission sensitivity (e.g., Brazil) inversion increments are
very small in all three cases and therefore the baseline choice has little impact. However, considering countries with higher emission sensitivities (e.g., China), the *a posteriori* emissions are very sensitive to the baseline definition. In almost all cases the REBS method leads to smaller and Stohl's method to larger national emissions than the GDB method. Due to the large

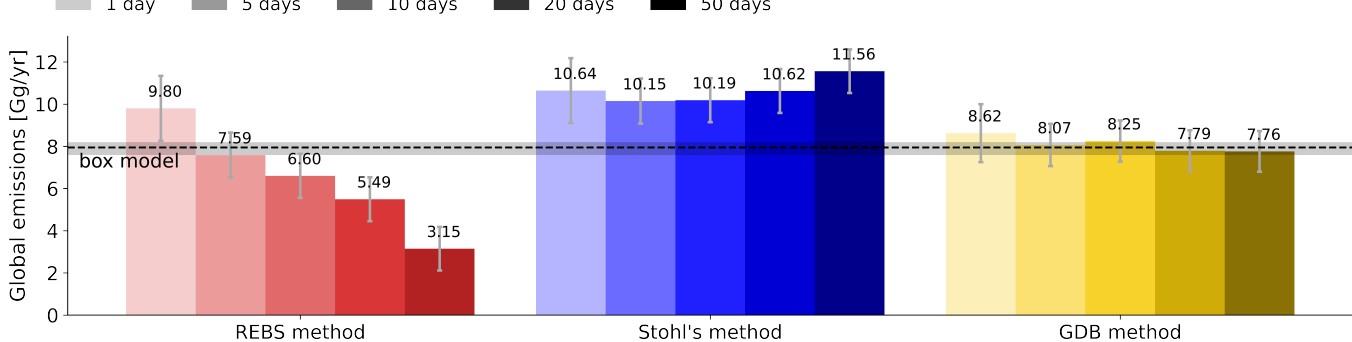

**Figure 10.** $SF_6$ global emissions derived by the inversions. Results are shown for the three applied baseline methods and for the five applied backward simulation periods between 1 and 50 days. The horizontal dashed line represents the reference value of the AGAGE 12-box model with shaded error bands. Uncertainties represent a $1\sigma$ range.

emissions in China the differences in *a posteriori* emissions become especially apparent there, with almost a factor 3 emission difference, corresponding to almost 30% of the 2012 global $SF_6$ emissions.

**Global emissions**

The 2012 $SF_6$ global emissions are shown in Fig. 10. The bars represent inversion results using different backward calculation periods between 1 and 50 days (light to dark shading). The horizontal dashed line illustrates a reference value calculated by Simmonds et al. (2020) with the AGAGE 12-box model. Notice that this is the same value used to calculate the *a priori* emissions, so the line represents also the global *a priori* emissions, which should be kept in mind for the interpretation of the results. Since the uncertainty of the global emissions is relatively small, global emissions derived by the inversion should roughly match the value of the box model, regardless which backward simulation period was used.

For the REBS method, calculated global emissions (red) decrease dramatically with growing backward simulation time, showing values between 3.15 Gg/yr and 9.80 Gg/yr. This is a consequence of the method's incapability to remove emission contributions from the baseline when the backward simulation period expands, leading to a systematical overestimation of the baseline and underestimation of the emissions. The resulting bias increases with growing simulation period and as a result global emissions estimates deviate strongly from the box model.

In the case of Stohl's method (blue), derived global emissions do not show such a systematic decrease with longer backward simulation periods as observed for the REBS method. This is because Stohl's method not only selects low mixing ratio observations, but also uses model information to maintain the baseline. For longer backward simulation periods, higher simulated mixing ratios are subtracted from the pre-selected observations to compensate for more direct emission contributions. Nevertheless, global emissions significantly exceed the reference value of the box model for all applied simulation periods, implying a systematic overestimation of emissions through too low baselines. The overestimation of the global emissions increases with longer backward simulation times larger than 5 days. This suggests that the method overcompensates for additional direct

emission contributions when the simulation period expands, subtracting systematically too high values from the pre-selected observations.

We further investigate whether the encountered biases can be reduced by optimizing the baseline in the inversion. Therefore, we repeated the inversion with exactly the same setup, except optimizing the REBS and Stohl's baseline as part of the inversion. Results are shown in Fig. A2. In case of the REBS method the baseline optimization has only little effect on the global total *a posteriori* emissions for backward simulation periods between 1 and 10 days and becomes noticeable only after 20 days. The greatest improvements can be observed for the 50-day simulation, where the bias is almost halved. Still, for longer simulation periods the increasing improvements through the baseline optimization cannot compensate for the growing underestimation of the emissions and substantial biases remain. Optimizing Stohl's baseline shows great improvements, especially for longer simulation periods. These improvements increase systematically with growing backward simulation period and results get very close to the box model outcome for the 20- and 50-day simulation case.

Considering the inversion results based on the GDB method, global emissions are in good agreement with the box model result for all tested backward simulation periods, as the global *a posteriori* emissions stay close to the global *a priori* value. Furthermore, these global emissions stay almost unchanged for different backward simulation periods, demonstrating the method's ability to adjust the baseline according to the sampled emissions of different simulation periods.

**The advantage of longer backward simulation periods**

As an argument for a relatively short backward simulation period Stohl et al. (2009) stated that "the value for the inversion of every additional simulation day decreases rapidly with time backward". Certainly, this is true for countries and regions that are well covered by the global monitoring network. For instance, for France the SRR increases rapidly in the first few backward simulation days but flattens to a linear increase for longer backward simulation periods (Fig. 11a). A similar behavior can be observed for many countries in the Northern Hemisphere, although the curve's slope for the first few days varies. For countries poorly covered by the monitoring network, however, the SRR is close to zero for the first 5 to 15 backward days and only longer backward simulations might provide information for the inversion (see Fig. 11b). For these countries, the SRR increase with time flattens to a linear increase only for very long transport times, even beyond the 50 days used in this study.

Figure 12 further illustrates the impact of different backward simulation periods on the inversion, by showing emission increments for the GDB method and for backward simulation periods of 1, 10 and 50 days. In case of 1-day backward calculations (Fig. 12a) the inversion significantly optimizes *a priori* emissions only in East Asia and parts of Europe. As the backward simulation period is extended to 10 days (Fig. 12b), the inversion optimizes emissions in larger parts of the Northern Hemisphere but in the Southern Hemisphere emission increments are still small. In case of 50 days (Fig. 12c), the inversion optimizes emissions even far away from observation stations (e.g. South America or South Africa). In India, where SRR values are also small and *a priori* emissions (and thus emission uncertainties) are high (see also 11b), the emission increments even switch from positive to negative by extending the period from 10 to 50 days. Also, the calculated relative uncertainty reduction increases by extending the backward simulation period (see Fig. A3a-c).

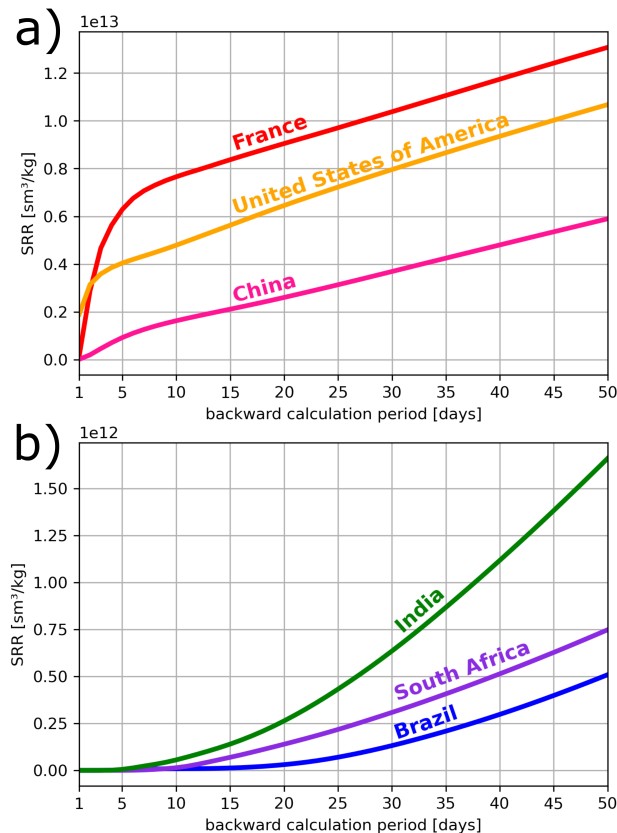

**Figure 11.** SRR for individual countries and different backward calculation periods between 1 to 50 days, considering all continuous measurement stations in Tab 1. The shown values are averages over the grid cells of a) France, USA, China and b) India, South Africa, Brazil, for the year 2012.

**The use of flask samples**

An advantage of the GDB method is the possibility to include flask measurements from fixed sites or moving platforms in the inversion. By contrast, the REBS and Stohl's method require short measurement intervals at fixed sites for the statistical baseline calculation. Here, the baseline could be taken from nearby or same latitude continuous sites, or represented through baselines at the domain border in case of regional inversions (Manning et al., 2021). Figure 13a shows the relative change in *a posteriori* emissions and Fig. 13b the additional relative error reduction when using flask measurements additionally to the continuous measurements for the 50-day backward simulation. One can see substantial differences in the USA, Eastern Europe, South Africa, East Asia and the Near East, where also an additional error reduction occurs. While this additional error reduction can be relatively large (up to 73%) for grid cells in the vicinity of the measurement sites, it quickly decreases down to a few percent with larger distance to the measurements. Consequently, flask measurements show only small influence on the total global emission estimate (< 1%), but can have a large impact on calculated national emissions of specific countries

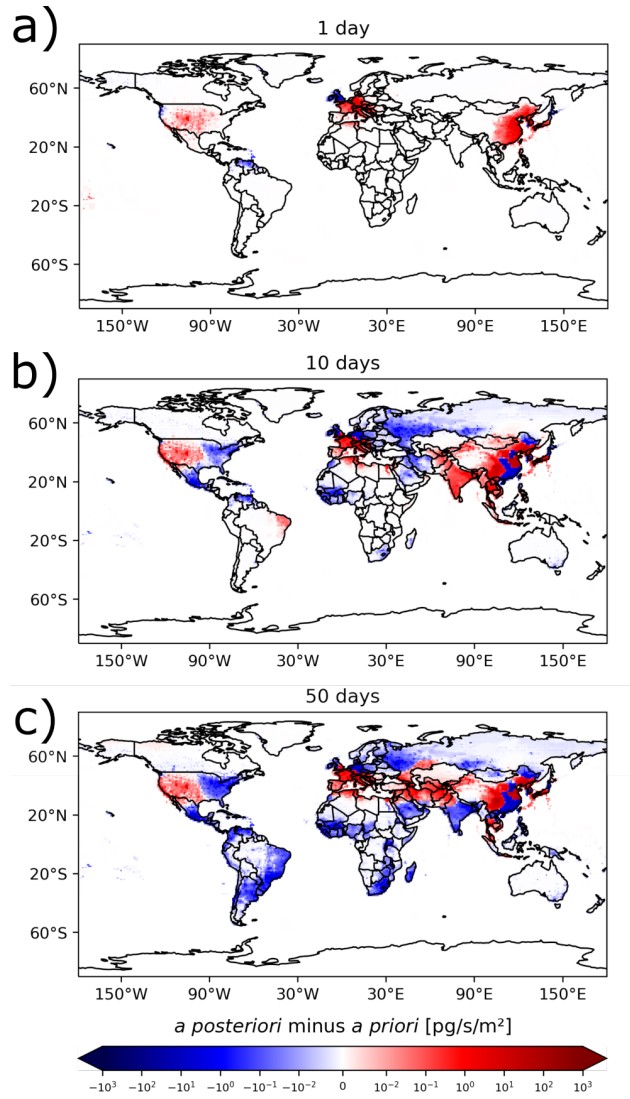

**Figure 12.** SF$_6$ emission increments calculated with the inversion by using the GDB method and a backward simulation period of a) 1 day, b) 10 days and c) 50 days

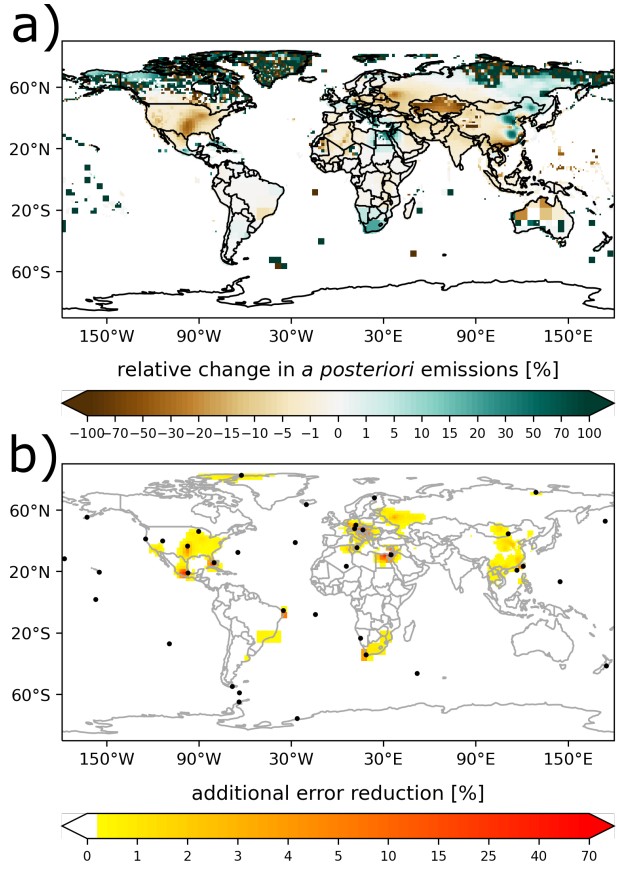

**Figure 13.** a) Relative change in *a posteriori* emissions and b) the additional error reduction when using flask measurements in addition to continuous measurements for the 50-day simulation. The locations of the flask measurements are marked with black dots.

(Fig. A4). For countries in the Near East the additional use of flask measurements changes national emission estimates by 40% to 100%. South African and American emissions are modified by around 10%.

**Reliable global emissions can only be obtained with long backward simulation periods**

In previous sections, we have used global mixing ratio fields from the GDB method where great care has been taken to avoid biases that would affect the baseline, and we have used global *a priori* emissions that correspond to the rather well known global $SF_6$ emissions. These are optimal conditions for the inversion that are rarely fulfilled for other species than $SF_6$. For many species global emissions are less well known, and with fewer observations than for $SF_6$ also the global distribution 505 (and, thus, the baseline) is more uncertain. However, a skillful inversion should tolerate such biases and still produce reliable results. While we lack information for verifying that regional emissions are reliable, for $SF_6$ we can at least test whether global emissions can be determined by our inversion in the presence of biases.

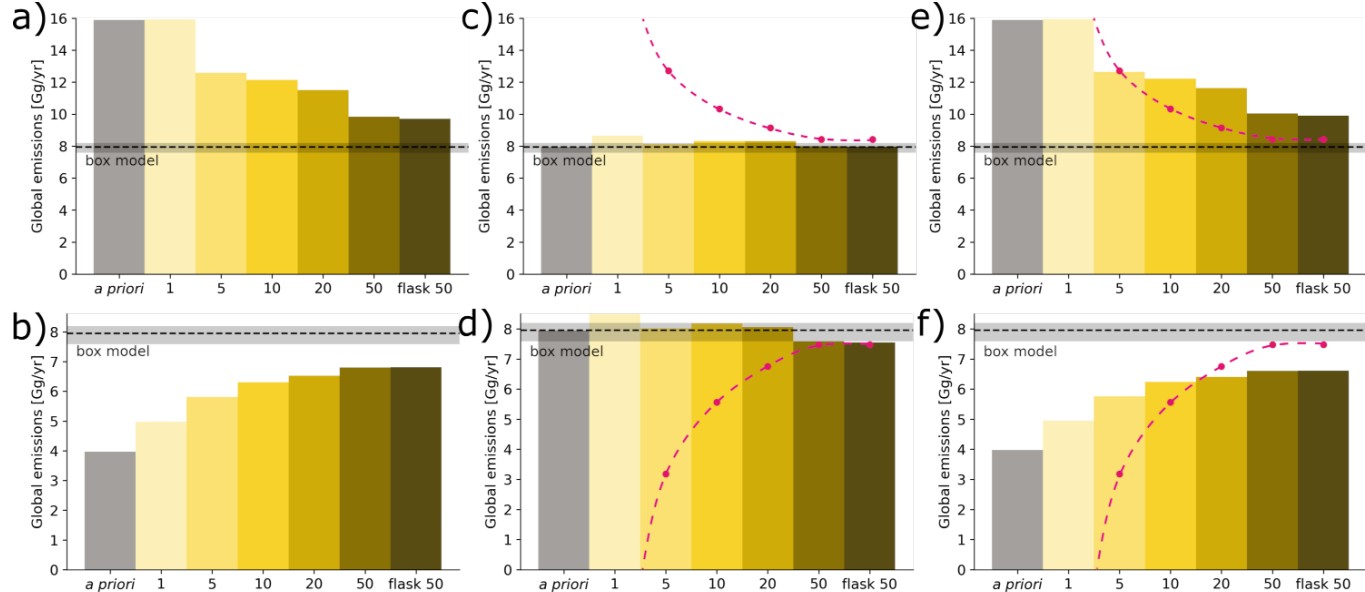

**Figure 14.** Global SF$_6$ emissions using the GDB method shown for different sensitivity cases, using backward simulation periods between 1 and 50 days, and a 50-day backward simulation case, where additionally to continuous measurements also flask measurements were included in the inversion. The sensitivity cases include a) doubled and b) halved *a priori* emissions; biased global mixing ratio fields with a uniform bias of c) -0.003 ppt and d) +0.003 ppt in every grid cell; and combinations of the two test types; e) doubled *a priori* emissions plus -0.003 ppt global field bias; f) halved *a priori* emissions plus +0.003 ppt global field bias.The dashed pink lines represent the expected relationship between the baseline bias and a resulting emission bias if a global box model was used and the bias attributed solely to emissions in different periods corresponding to the backward simulation times.

Figure 14 shows global *a posteriori* emissions when biases in (1) the *a priori* emissions and (2) global mixing ratio fields were added. This is shown for different backward simulation periods between 1 and 50 days and for the 50-day case with the inclusion of flask measurements. Note that for all these sensitivity cases shown in Fig. 14 we use the same absolute *a priori* emission uncertainties, as for the original *a priori* emissions without any artificial bias.

Comparing the inversion results for doubled (Fig. 14a) and halved (Fig. 14b) *a priori* emissions clearly shows that the corresponding biases in the global *a posteriori* emissions are reduced substantially with increasing backward simulation period and converge towards the rather well known global SF$_6$ emission from the box model. However, it is clear that a substantial bias remains even with a backward simulation period of 50 days. It seems likely that an extension of the backward simulation period beyond 50 days would further reduce the bias. The inclusion of flask measurements leads to slight additional improvements.

Another sensitivity test was performed with artificially biased global mixing ratio fields by subtracting (Fig. 14c) or adding (Fig. 14d) 0.003 ppt from/to the FLEXPART CTM model output in every grid cell of the 3D mixing ratio fields. 0.003 ppt is equivalent to roughly 1% of the 2012 global mixing ratio increase and thus corresponds to about 3 days of global SF$_6$ emissions. To still fit the model to the observations, the inversion will try to compensate such a bias in the baseline with a

bias of the opposite sign in the emissions. As always, the inversion can only attribute this additional bias to emissions within the simulation period. Therefore, shorter backward simulation periods require a greater modification of emissions than longer periods, in order to compensate for the baseline bias. To fully compensate the baseline bias equivalent to 3 days of emissions, global a *posteriori* emissions would need to deviate strongly from the reference value for the 1-day case, but converge towards it with increasing backward simulation time. This is shown by the dashed pink line, which indicates the expected relationship between this baseline bias and a resulting emission bias if a global box model was used and the bias attributed solely to emissions in different periods corresponding to the backward simulation times. In fact, with a positive baseline bias negative emissions would be required for backward simulation times of less than 3 days, as the baseline exceeds the observations. The inversion results do not show this extreme behavior, since for short backward simulation times high SRR values are found only in small regions, and the emission changes there are bound by the prescribed *a priori* uncertainties. Notice that in our case of a known added bias, this is rather a shortcoming, as this shows that the inversion is not able to compensate the baseline bias for short backward simulation times. Only for the longest times, the emissions converge towards the expected global emissions (dashed pink lines), and only for such long backward simulation times baseline biases equivalent to 3 days of emissions become negligible. We also investigated the inversion behavior for larger baseline biases, subtracting/adding (Fig. A5a/b) 0.05 ppt from/to the global fields, corresponding to roughly 50 days of the 2012 global $SF_6$ emissions. Here again, the results for short simulation times seem unpredictable, i.e. they do not follow the described expected behavior, indicated by the pink dashed lines. Only for the 50-days simulation periods results converge to the expected global emissions consistent with the respective baseline bias.

Finally, we also combined doubled *a priori* emissions with a -0.003 ppt bias in the global mixing ratio fields (Fig. 14e) and halved *a priori* emissions with a +0.003 ppt bias (Fig. 14f). For both cases, the inversion becomes less sensitive to biases in the *a priori* emissions and the global fields with longer backward simulation periods.

**Final remark**

In this study, we show many advantages for using relatively long backward simulation periods for the inversion. Nevertheless, the improvement of regional emission patterns is still limited by the observation network. A lack of observations in one region cannot simply be compensated by extending the simulations for stations in other regions to very long periods. For backward simulation times of 20-50 days, the emission sensitivity is distributed over large areas but usually still concentrated within broad latitude bands. The additional information to be gained from such long simulation times, on top of the information provided by the shorter simulation times, can probably best be compared with the inversions done with a multi-box model such as the AGAGE 8-box model (e.g. Rigby et al., 2013) that is capable of determining the emissions in broad latitude bands. Consequently, if the emissions in certain regions with a dense observation network are already well constrained by shorter simulation periods, the residual emission will be attributed correctly as an emission total to all other regions of the same latitude band with a poor station coverage. The effective resolution of the obtained emissions in such data-poor regions may be very coarse but the result might still be informative. Furthermore, the emission sensitivity for the 20-50 day backward period is still not uniformly distributed over a latitude band and thus provides some limited regional information. Perhaps

supported with a limited number of strategically located flask measurements, inversions with long backward simulation times could provide coarse but robust information on emissions in poorly sampled regions. Independently, the growing correlation between modeled and observed mixing ratios with increasing backward simulation length (Tab. 2; averaged over all stations) also shows that longer backward simulations hold additional information, even though the information gain decreases with every day added to the simulation length and probably becomes marginal for very long backward simulation times. However, we propose to make use of this additional information and apply longer periods whenever possible to make the best use of the existing observation network.

## 4   Conclusions

We have examined the use of LPDMs for inverse modeling of GHG emissions by varying the backward simulation period in the range of 1 to 50 days, testing several methods for estimating a baseline, investigating the influence of biases in the *a priori* emissions and the baseline, and exploring the value of flask measurements for the inversion. We found the following:

– A baseline method that is purely based on the observations at the observation site itself, such as the REBS method, may lead to unreliable inversion results that are highly sensitive to the length of the LPDM backward simulation and can lead to unrealistic *a posteriori* global total emissions. For instance, for the year 2012 inversions with the REBS method produce *a posteriori* global total $SF_6$ emissions ranging between 9.8 Gg/yr and 3.2 Gg/yr for backward simulation periods between 1 day and 50 days, compared to a well known reference value of around 8.0 Gg/yr. Optimizing the baseline shows little effect for simulation periods between 1 and 20 days, but could half the bias in the 50-day simulation case. Although the improvements of the baseline optimization increase with growing backward simulation period, the simultaneously growing bias cannot be compensated.

– A baseline method that is based on the observations at the site itself but corrects for emissions occurring during the LPDM backward simulation period leads to smaller sensitivity to the backward calculation time but may still lead to substantially biased emissions irrespective of backward simulation period. For instance, inversions with Stohl's method overestimate the well known 2012 $SF_6$ global total emissions by 2.2 - 3.6 Gg/yr (28-45%). Optimizing the baseline, however, shows great improvements, especially for longer simulation periods.

– A global distribution based (GDB) approach, where the LPDM backward simulation is nested into a global mixing ratio field, leads to *a posteriori* emissions that are less sensitive to LPDM backward calculation lengths and stay close to the global total emission value. In contrast to station-specific baselines, the GDB method allows the inclusion of low-frequency measurements (e.g., flask samples) or data from mobile platforms into the inversion.

– Statistical comparisons of *a priori* modeled versus observed mixing ratios suggest that longer LPDM backward simulations outperform shorter simulations. In particular, extending the trajectory length from 5-10 days to 50 days can reduce the mean squared error and increase the correlation.

- Inverse modeling is highly sensitive to biases in the *a priori* emissions as well as biases in the baseline. We could show that this sensitivity can decrease with the length of the backward simulation period and we find that longer backward simulation periods can help to correct biased global emission fields. In the presented case, it is not possible to correct strongly biased global *a priori* emissions with backward simulation periods of 1-10 days, while they are captured quite accurately with 50-day backward simulations.

- The additional use of flask measurements has the potential to improve the observational constraint on $SF_6$ emissions, especially close to the measurement sites. However, existing flask sampling sites are often not well located for inversion purposes. Similar to Weiss et al. (2021) we suggest that placing a few additional flask sampling sites downwind of potential emission regions in currently undersampled parts of the world (in particular, tropical South America, tropical Africa, India, Australia and the Maritime Continent) would have disproportionately large value for improving regional and global *a posteriori* emission fields.

Following these results, we advise against the use of baseline methods that are purely based on the observations of individual sites. At least great care needs to be taken that problems such as demonstrated in this paper do not occur. In order to reduce biases, the optimization of the baseline as part of the inversion might be necessary, but would likely not be sufficient to avoid biases completely. We recommend also to employ longer LPDM backward simulation periods, beyond 5-10 days, as this can lead to improvements in overall model performance, can produce more robust global emission estimates and might help to constrain emissions, at least at very coarse resolution, in regions poorly covered by the monitoring network. When consistency between regional and global emission estimates is important, even longer backward simulation periods than 50 days may be useful. Finally, we suggest to take additional flask measurements at continental sites in the Tropics and Southern Hemisphere as they would greatly enhance inversion-derived global emission fields.

*Code and data availability.* The used source codes of FLEXPART 10.4 and FLEXINVERT+ (with small modifications to the original version freely available at https://flexinvert.nilu.no/download.html; downloaded in July 2020; (described in detail by Thompson and Stohl, 2014)) are provided at https://doi.org/10.25365/phaidra.339, together with input-, setting- and output data. The source code of FLEXPART 8-CTM-1.1 together with an users guide can be freely downloaded at https://doi.org/10.5281/zenodo.1249190. The source code of FLEXPART 10.4 is also freely available on the FLEXPART website: https://www.flexpart.eu/downloads (described in detail by Pisso et al., 2019). Atmospheric measurements of $SF_6$ mixing ratios used in this study are freely available from the following sources: AGAGE data: http://agage.eas.gatech.edu/data_archive/, NOAA ESRL data: https://gml.noaa.gov/dv/data/, NOAA Carbon Cycle Group ObsPack data: https://doi.org/10.25925/20180817, World Data Centre for Greenhouse Gase: https://gaw.kishou.go.jp/. All the listed websites were last accessed on 27.04.2022.

*Author contributions.* MV and AS designed the study with contributions from RT. MV performed the FLEXPART, FLEXPART CTM, and FLEXINVERT+ simulations. RT helped with the FLEXINVERT+ setup and simulation issues. MV made the figures with help from AP. MV wrote the text with input from AS, AP, and RT.

*Competing interests.* The authors declare that they have no conflict of interest.

*Acknowledgements.* We thank the whole AGAGE team for providing measurement data, including J. Mühle (Scripps Institution of Oceanography); P.B. Krummel, P.J. Fraser, L.P. Steele (CSIRO Oceans and Atmosphere); R. Wang (Georgia Institute of Technology); S. O'Doherty, D. Young (University of Bristol); M.K. Vollmer, S. Reimann (EMPA: Swiss Federal Laboratories for Materials Science and Technology); C.R. Lunder, O. Hermanson (NILU: Norwegian Institute for Air Research). AGAGE operations at Mace Head, Trinidad Head, Cape Matatula, Ragged Point, and Cape Grim are supported by NASA (USA) grants to MIT (NAG5-12669, NNX07AE89G, NNX11AF17G, NNX16AC98G) and SIO (NNX07AE87G, NNX07AF09G, NNX11AF15G, NNX11AF16G, NNX16AC96G, NNX16AC97G), and also by: Department for Business, Energy & Industrial Strategy (BEIS, UK), Contract 1537/06/2018 to the University of Bristol for Mace Head and NOAA (USA), contract 1305M319CNRMJ0028 to the University of Bristol for Ragged Point; CSIRO and BoM (Australia): FOEN grants to Empa (Switzerland); NILU (Norway); SNU (Korea); CMA (China); NIES (Japan); and Urbino University (Italy). For Jungfraujoch funding is acknowledged for the project HALCLIM/CLIMGAS-CH by the Swiss Federal Office for the Environment (FOEN) and for ICOS (Integrated Carbon Observation System) by the Swiss National Science Foundation. In addition, measurements are supported by the International Foundation High Altitude Research Stations Jungfraujoch and Gornergrat (HFSJG). The Commonwealth Scientific and Industrial Research Organisation (CSIRO, Australia) and Bureau of Meteorology (Australia) are thanked for their ongoing long-term support and funding of the Cape Grim station and the Cape Grim science program. We also thank the NOAA Global Monitoring Laboratory for providing access to their data, including G. Dutton, J.W. Elkins, B. Hall, C. Sweeney, E. Dlugokencky, A. Andrews, D. Nance, and keypartners: L. Huang (EC), K. Davis (PSU), S. Biraud (LBNL-ARM). We further acknowledge following people and institutions for sharing their observation data: T. Saito (National Institute for Environmental Studies, Japan); S. Park, M. Park (Kyungpook National University - operations of the Gosan station on Jeju Island, South Korea were supported by the National Research Foundation of Korea grant funded by the Korean government MSIT no. 2020R1A2C3003774); E. Cuevas (State Meteorological Agency, Spain); D.Say (University of Bristol). We also thank C. Groot Zwaaftink, S. Eckhardt (NILU), and S. Henne (EMPA) for providing and supporting with the Flexpart CTM model. Further acknowledgement is made for the use of ECMWF's computing and archive facilities provided through a special project (spatvojt) in this research. We further thank Marina Dütsch, Lucie Bakels, Silvia Bucci, Katharina Baier, Daria Tatsii, and Perta Seibert for their support.

**Table A1.** Surface flask measurement sites

| Site ID | Station | Latitude | Longitude | Altitude[a] |
|---------|---------|----------|-----------|-------------|
| ALT | Alert, Canada | 82.5°N | 62.5°W | 190 |
| ASC | Ascension Island, UK | 8.0°S | 14.4°W | 90 |
| ASK | Assekrem, Algeria | 23.3°N | 5.6°E | 2715 |
| AZR | Serreta (Terceira), Portugal | 38.8°N | 27.4°W | 24 |
| BAL | Baltic Sea, Poland | 55.4°N | 17.2°E | 28 |
| BHD | Baring Head, New Zealand | 41.4°S | 174.9°E | 90 |
| BKT | Bukit Kototabang, Indonesia | 0.2°S | 100.3°E | 875 |
| BMW | Tudor Hill (Bermuda), UK | 32.3°N | 64.9°W | 60 |
| BSC | Constanta (Black Sea), Romania | 44.2°N | 28.7°E | 5 |
| CBA | Cold Bay (AK), USA | 55.2°N | 162.7°W | 57 |
| CHR | Christmas Island, Kiribati | 1.7°N | 157.2°W | 5 |
| CPT | Cape Point, South Africa | 34.4°S | 18.5°E | 260 |
| CRZ | Crozet, France | 46.4°S | 51.8°E | 202 |
| DRP | Drake Passage, USA | 59.0°S | 63.7°W | 10 |
| DSI | Dongsha Island, Taiwan | 20.7°N | 116.7°E | 8 |
| EIC | Easter Island, Chile | 27.2°S | 109.4°W | 69 |
| GMI | Guam (Mariana Island), USA | 13.4°N | 144.7°E | 5 |
| HBA | Halley, UK | 75.6°S | 26.2°W | 35 |
| HFM | Harvard Forest (MA), USA | 42.5°N | 72.2°W | 1000 |
| HPB | Hohenpeissenberg, Germany | 47.8°N | 11.0°E | 941 |
| HSU | Humboldt State University, USA | 41.0°N | 124.7°W | 8 |
| HUN | Hegyhatsal, Hungary | 47.0°N | 16.6°E | 344 |
| ICE | Storhofdi, Iceland | 63.4°N | 20.3°W | 127 |
| KEY | Key Biscane (FL), USA | 25.7°N | 80.2°W | 6 |
| KUM | Cape Kumukahi (HI), USA | 19.5°N | 154.8°W | 8 |
| LEF | Park Falls (WI), USA | 45.9°N | 90.3°W | 868 |
| LLN | Lulin, Taiwan, Province of China | 23.5°N | 120.9°E | 2867 |
| LMP | Lampedusa, Italy | 35.5°N | 12.6°E | 50 |
| MEX | Mex High Altitude Global Climate Observation Center, Mexico | 19.0°N | 97.3°W | 4469 |
| MID | Sand Island, USA | 28.2°N | 177.4°W | 16 |
| MKN | Mt. Kenya, Kenya | 0.1°S | 37.3°E | 3649 |
| NAT | Natal, Brazil | 5.5°S | 35.3°W | 20 |
| NMB | Gobabeb, Namibia | 23.6°S | 15.0°E | 461 |
| OXK | Ochsenkopf, Germany | 50.0°N | 11.8°E | 1172 |
| PAL | Pallas, Finland | 68.0°N | 24.1°E | 570 |
| PSA | Palmer Station, USA | 64.9°S | 64.0°W | 15 |
| PTA | Point Arena (CA), USA | 39.0°N | 123.7°W | 22 |
| SGP | Southern Great Plains E13 (OK), USA | 36.6°N | 97.5°W | 374 |
| SHM | Shemya Island, USA | 52.7°N | 174.1°E | 28 |
| TIK | Tiksi, Russian Federation | 71.6°N | 128.9°E | 29 |
| USH | Ushuaia, Argentina | 54.8°S | 68.3°W | 32 |
| UTA | Wendover (UT), USA | 39.9°N | 113.7°W | 1332 |
| UUM | Ulaan Uul, Mongolia | 44.5°N | 111.1°E | 1012 |
| WIS | Sede Boker, Israel | 30.9°N | 34.8°E | 482 |

[a] The altitude specifies the sampling height in meters above sea level.

**Table A2.** Aircraft flask measurement programs

| Site ID | Aircraft Programs | Latitude | Longitude | Altitude[a] |
|---------|-------------------|----------|-----------|-------------|
| BNE | Beaver Crossing, Nebraska, United States | 40.8°N | 97.2°W | 616 - 7855 |
| CAR | Briggsdale, Colorado, United States | 40.7°N | 104.3°W | 1795 - 8469 |
| CMA | Cape May, New Jersey, United States | 38.9°N | 74.3°W | 280 - 8010 |
| DND | Dahlen, North Dakota, United States | 47.5°N | 99.1°W | 587 - 8023 |
| ESP | Estevan Point, British Columbia, Canada | 49.4°N | 126.4°W | 246 - 5740 |
| ETL | East Trout Lake, Saskatchewan, Canada | 54.3°N | 104.9°W | 811 - 7823 |
| HIL | Homer, Illinois, United States | 40.0°N | 87.9°W | 555 - 8051 |
| LEF | Park Falls, Wisconsin, United States | 46.0°N | 90.2°W | 583 - 4018 |
| NHA | Worcester, Massachusetts, United States | 42.9°N | 70.5°W | 245 - 8069 |
| PFA | Poker Flat, Alaska, United States | 64.8°N | 148.2°W | 222 - 7444 |
| RTA | Rarotonga, Cook Islands | 21.2°S | 159.8°W | 15 - 6483 |
| SCA | Charleston, South Carolina, United States | 32.9°N | 79.5°W | 218 - 8070 |
| SGP | Southern Great Plains, Oklahoma, United States | 36.6°N | 97.5°W | 437 - 5716 |
| TGC | Sinton, Texas, United States | 27.7°N | 96.7°W | 250 - 8074 |
| THD | Trinidad Head, California, United States | 41.1°N | 124.2°W | 231 - 8034 |
| WBI | West Branch, Iowa, United States | 41.7°N | 91.3°W | 591 - 8204 |

[a] The altitude specifies the range of sampling heights in meters above sea level.

**Table A3.** Setting parameters of the REBS method. For more information see Ruckstuhl et al. (2012).

| Setting Parameters | Description |
| --- | --- |
| b = 2.5 | tuning factor which governs the weight of outliers in the baseline |
| $span = \frac{1}{6}$ | the ratio of observation points used to compute one baseline value. (the goal is a temporal window of 2 months). It regulates the amount of baseline smoothing. |
| maxit = c(10, 10) | maximum number of iterations |

**Table A4.** Nudging kernel settings for surface and aircraft measurement sites. The kernels are set to have an equal spatial length (in meters) in the x and the y direction. For surface measurement sites in the northern hemisphere, an upper limit for $h_y$ was set to 25°; $\sigma_{obs}$ defines the standard deviation of measurements over the simulation period at each nudging location; $\sigma_{max}$ describes the maximum value of $\sigma_{obs}$ from all surface observation stations. For aircraft measurement sites, the kernel size depends on the height level above ground H. For additional information on the parameters see Groot Zwaaftink et al. (2018)

| surface measurements sites | | | | |
|---|---|---|---|---|
| hemisphere | spatial kernel width $h_y$ [°] | kernel height $h_z$ [m] | temporal kernel length $h_t$ [s] | kernel relaxation time $\tau$ [s] |
| northern hemisphere | $h_y = \frac{\sigma_{max}}{\sigma_{obs}} \cdot 2$ | $h_z = 300$ | $h_t = 86{,}400 \cdot \frac{\sigma_{max}}{\sigma_{obs}}$ | $\tau = 3600$ |
| southern hemisphere | $h_y = 25$ | $hz = 500$ | $h_t = 86{,}400 \cdot \frac{\sigma_{max}}{\sigma_{obs}}$ | $\tau = 3600$ |
| aircraft measurements sites | | | | |
| height H [km above ground] | spatial kernel width $h_y$ [°] | kernel height $h_z$ [m] | temporal kernel length $h_t$ [s] | kernel relaxation time $\tau$ [s] |
| $H \leqslant 0.5$ | $h_y = 10$ | $h_z = 100$ | $h_t = 86{,}400 \cdot \frac{h_y}{1°}$ | $\tau = 3600$ |
| $0.5 < H \leqslant 1$ | $h_y = 20$ | $h_z = 250$ | $h_t = 86{,}400 \cdot \frac{h_y}{1°}$ | $\tau = 3600$ |
| $1 < H \leqslant 2$ | $h_y = 30$ | $h_z = 500$ | $h_t = 86{,}400 \cdot \frac{h_y}{1°}$ | $\tau = 3600$ |
| $2 < H \leqslant 3$ | $h_y = 40$ | $h_z = 500$ | $h_t = 86{,}400 \cdot \frac{h_y}{1°}$ | $\tau = 3600$ |
| $3 < H \leqslant 4$ | $h_y = 50$ | $h_z = 500$ | $h_t = 86{,}400 \cdot \frac{h_y}{1°}$ | $\tau = 3600$ |
| $4 < H \leqslant 5$ | $h_y = 60$ | $h_z = 500$ | $h_t = 86{,}400 \cdot \frac{h_y}{1°}$ | $\tau = 3600$ |
| $5 < H \leqslant 6$ | $h_y = 70$ | $h_z = 1000$ | $h_t = 86{,}400 \cdot \frac{h_y}{1°}$ | $\tau = 3600$ |
| $6 < H \leqslant 7$ | $h_y = 80$ | $h_z = 1000$ | $h_t = 86{,}400 \cdot \frac{h_y}{1°}$ | $\tau = 3600$ |
| $7 < H \leqslant 8$ | $h_y = 90$ | $h_z = 1000$ | $h_t = 86{,}400 \cdot \frac{h_y}{1°}$ | $\tau = 3600$ |
| $8 < H \leqslant 9$ | $h_y = 100$ | $h_z = 1000$ | $h_t = 86{,}400 \cdot \frac{h_y}{1°}$ | $\tau = 3600$ |

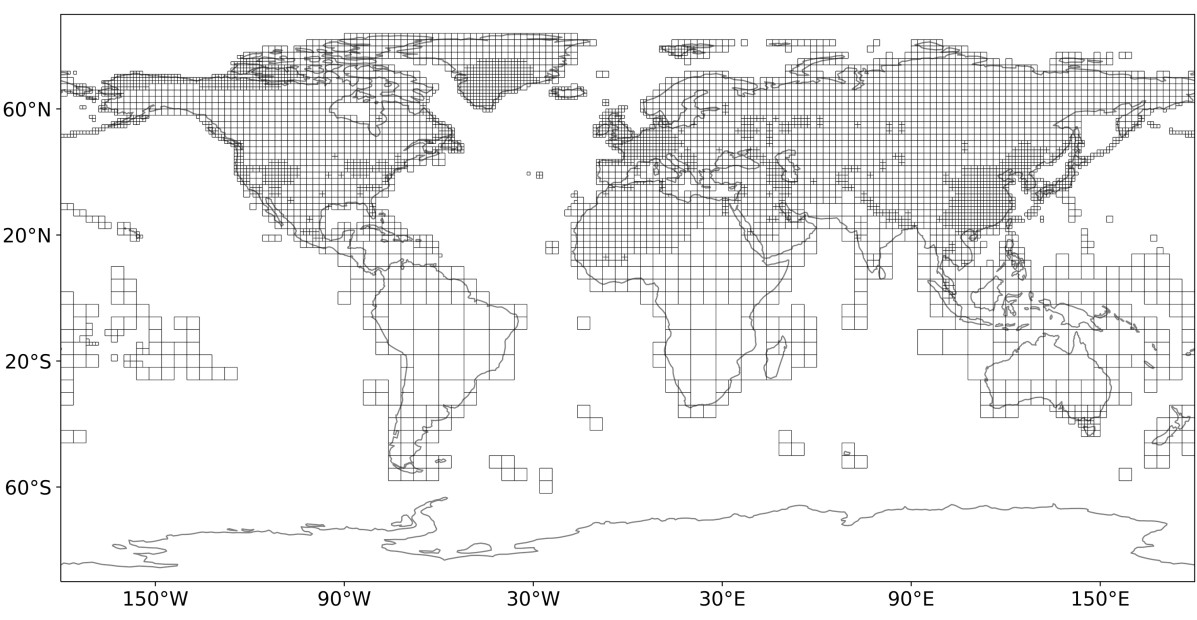

**Figure A1.** Variable-resolution grid on which emissions are optimized by the inversion.

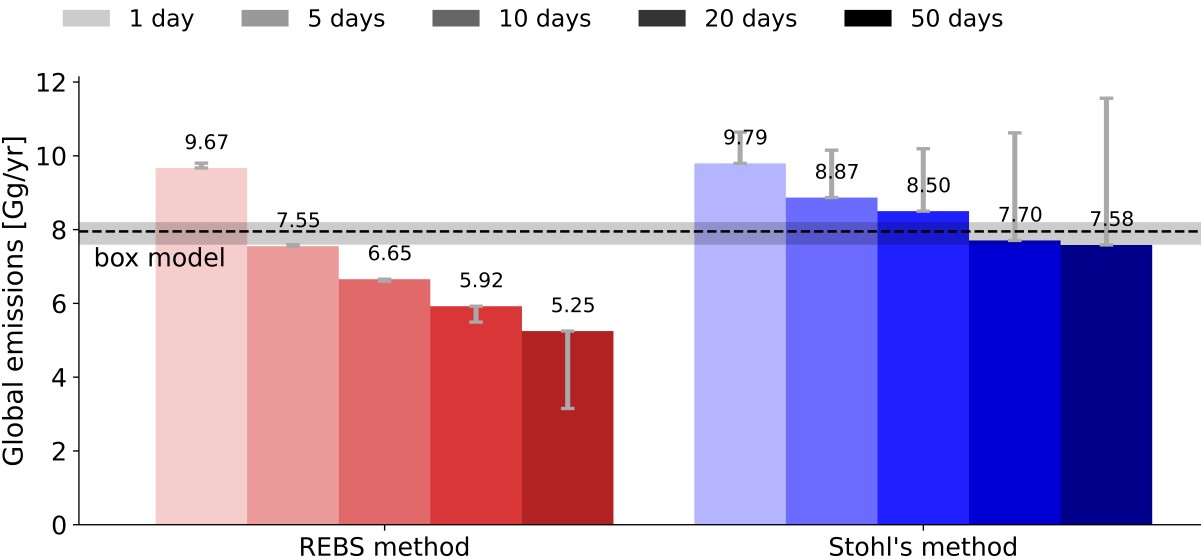

**Figure A2.** Calculated $SF_6$ global emissions when baseline concentrations are optimized as part of the inversion. Grey bars represent the improvements obtained by the baseline optimization. Results are shown for the REBS and Stohl's method and for all five applied simulation periods between 1 and 50 days. The horizontal dashed line represents the reference value of the AGAGE 12-box model with shaded error bands.

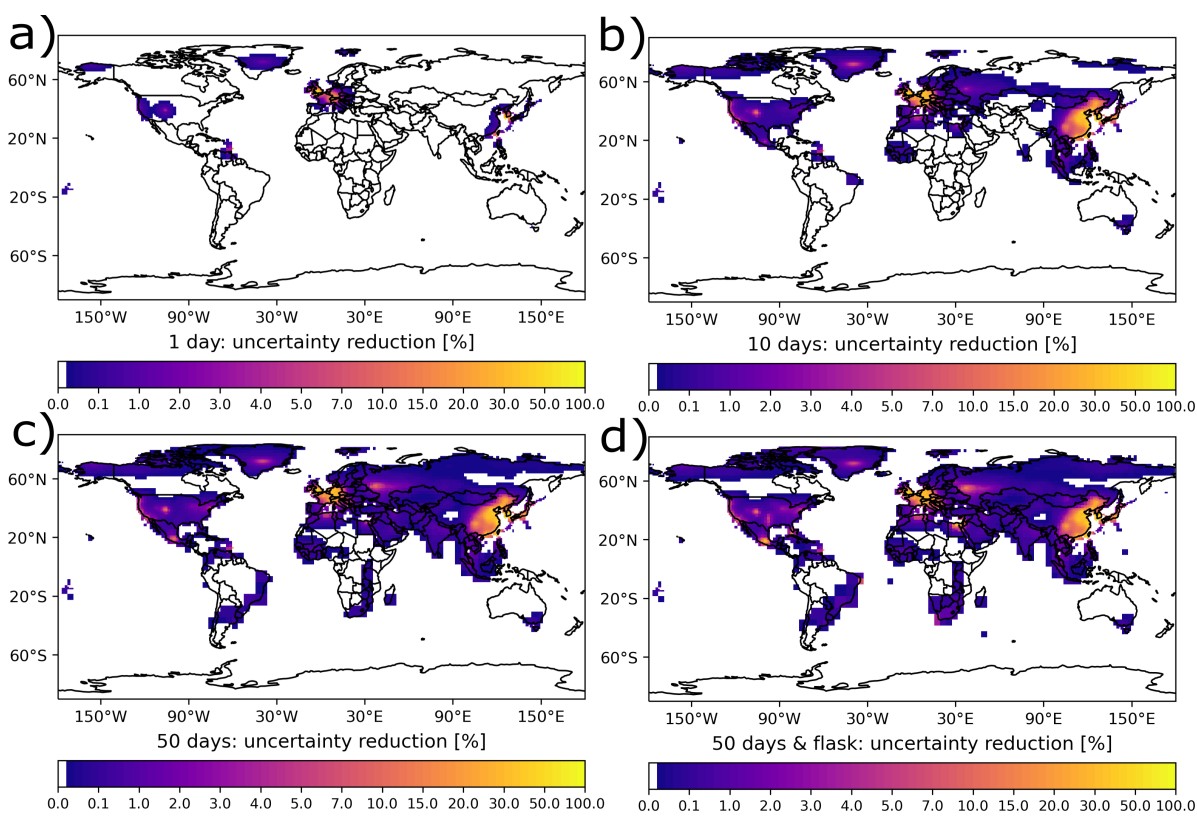

**Figure A3.** Relative uncertainty reductions $(1 - \frac{u_{post}}{u_{pri}})$ calculated with the inversion by using the GDB method and a backward simulation period for a) 1 day, b) 10 days, c) 50 days, and d) for the 50-day case, where also flask measurements were included

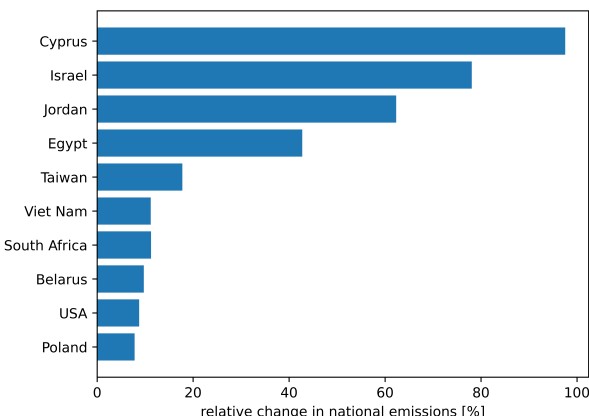

**Figure A4.** Relative change in national *a posteriori* emissions of selected countries, when flask measurements are used in addition to continuous measurements in the case of 50-day simulations.

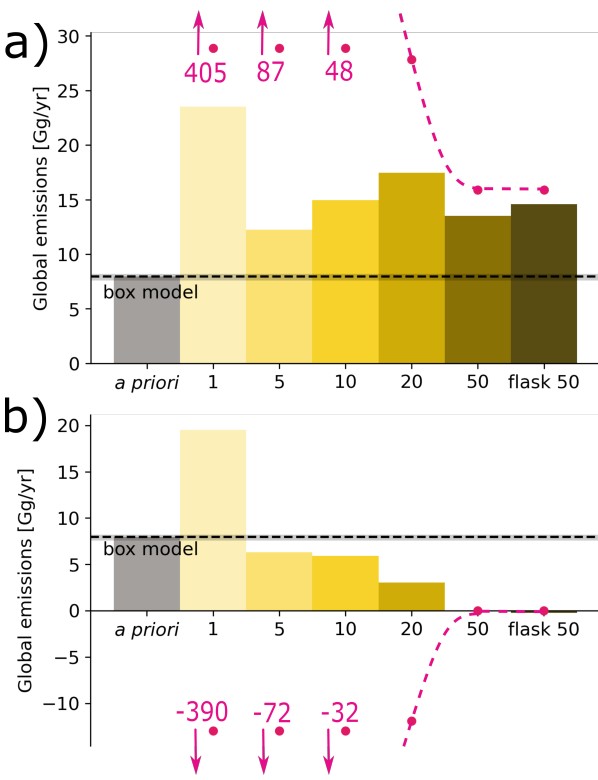

**Figure A5.** Global SF$_6$ emissions using the GDB method shown for two sensitivity tests, where a uniform bias of c) -0.05 ppt and d) +0.05 ppt is added to every grid cell of the global mixing ratio fields. Results are shown for backward simulation periods between 1 and 50 days, and for a 50-days backward simulation case, where additionally to continuous measurements also flask measurements were included in the inversion. The dashed pink lines represent the expected relationship between the baseline bias and a resulting emission bias if a global box model was used and the bias attributed solely to emissions in different periods. For these two sensitivity tests, *a priori* uncertainties were set to 500%.

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
