# Peer review of "A comprehensive evaluation of the use of Lagrangian particle dispersion models for inverse modeling of greenhouse gas emissions"

_EGUsphere, 2022_

## Author Comment (AC1)

**Author's response #1**

This paper presents a study into two factors that influence inverse estimates of greenhouse gas emissions using Lagrangian particle dispersion models (LPDMs): the period over which the models are run backwards in time, and the choice of "baseline" estimation method. I think this paper is suitable for publication in GMD. However, I think greater care needs to be taken when attempting to generalise some results.

My main criticism of the paper is around the way that the use of backward simulation time is discussed. It is no doubt true that very short simulation lengths will likely "miss" important influence on observations of nearby sources. However, there will be substantial diminishing returns for very long simulation periods, because of the rapid decline in sensitivity and increase in model uncertainty with distance from the measurement stations (and hence simulation time). Therefore, I think the authors need to be careful not to imply that we can solve for very far-field emissions using a very sparse measurement network and "long-enough" simulation periods. To provide some context, a recent study (Rigby et al., 2019) that used Gosan and Hateruma data suggested that robust emissions estimates of CFC-11 could only be derived for the eastern provinces of China (10 and 30-day simulations were used), and that as additional provinces were added further west, a posteriori uncertainty increased dramatically (and I believe the level of agreement between the four different methods decreased). In this paper, it is implied that, using essentially the same network in Asia, we could infer SF6 emissions from, for example, India (thus extending the domain from ~100s of km to ~1000s of km) by going from 5 to 50-day simulations. While there is of course some sensitivity to these distant regions, in practice, I very strongly doubt whether robust emission estimates could be derived so far from the stations, because of the very small SRR (around two orders of magnitude difference between India and eastern China, according to Figure 2) and comparatively high model uncertainty for such long trajectories.

Building on this point, I believe that one of the reasons this framing is arrived at is because the GDB method gives consistent results for different simulation lengths (Figure 10). However, I assume that the reason for this invariance to simulation length is because the inversion does not significantly deviate from the prior in regions further from the measurement station (at least when integrated over large scales). In the case of very short simulations, the sensitivity is necessarily only to emissions very close to the measurement stations. But even in the case of very long simulations, the SRR is so small in far regions, compared to the near-by areas, that it doesn't really make much difference at the global scale. Indeed, I think that this explains the persistent bias for all simulation lengths when the a priori fluxes are changed in Figure 14 (i.e., the low sensitivity prevents the inversion from overcoming the bias in the prior). I do think the tests show convincingly that the GDB method does a better job of compensating for issues relating to simulation length than the other baseline methods. Therefore, I suggest the authors take care that the discussion emphasises this outcome, without implying that more can be drawn from very long simulations than is possible in practice. Further elaboration and specific suggestions are provided below.

We would like to thank reviewer #1 for the valuable and constructive review of our manuscript.

In the response we use 4 different colors. The blue colored text is the general answer to the reviewer's comments. Additionally, we show how the text is changed in the manuscript: The original text is colored grey, removed text is colored red, and new text is colored green.

We acknowledge that the discussion around the backward simulation period needs to be handled with more care. We totally agree with the fact that a lack of observations in a sparse measurement network cannot be compensated through "long enough" simulation periods and it is not our intention to imply that. It is correct that the SRR values get more widespread and thus less specific with every additional day of backward calculation and therefore the emission patterns in certain regions far away from the observations network cannot be determined accurately, even with relatively long simulation periods. At the same time, it is clear that one should make the best use of the available observation data, and we suggest that there is still valuable information contained in the SRR values for backward simulation times longer than 5-10 days, which increases the value of the available data. We support this claim with three facts:

1) The correlations between modeled and observed mixing ratios (Table 2; averaged over all stations) continuously increase with increasing backward time. The increases in the explained variance per day backward becomes smaller and smaller with time, so is incremental but can be noticed even up to 50 days. Thus, to make the best use of the existing observations, longer backward simulations are beneficial, even though the gain in information decreases with every day backward. As long as the costs for longer backward calculations are small compared to the costs involved in performing the measurements, the diminishing return in improved accuracy for every extra day of calculation should not discourage us from making longer simulations.

2) Longer backward simulations improve global emission estimates as shown in Fig. 14, even though the improvement of regional emission patterns will be limited. Regarding Fig. 14, we think that an extension of the backward simulation period beyond 50 days would further reduce the bias of the retrieved global emissions. Imagine the extreme case of multi-year-long simulations: particles will be equally distributed around the globe and the total global emissions could be derived similar to a box model - even with only one station. Thus, an inversion based on very long backward simulation periods should always give a quite accurate global emission, presumably with a better accuracy than a simple global box model - at least for species with lifetimes of decades or more. Inversions with shorter backward simulation times, in contrast, will usually result in erroneous global emissions. Fig. 14 confirms this behavior for up to 50 days but we have no doubt that even longer simulations would give even more accurate global emissions than we have obtained with our 50-day simulations, especially when starting with a "wrong" prior.

3) We argue further that by getting a better constrain on the global emissions, and on regions which are well covered by observations, we also get improved emission estimates of poorly sampled regions, however, without resolving the exact spatial emission patterns in these poorly sampled regions. We think that emissions can be improved as long as the growth in SRR is above linear (Fig. 11), which for some regions will happen only after a very long simulation period. As shown in Fig. 12, the inversion produces non-zero emission increments at regions, that were untouched for smaller simulation periods. We would like to repeat that we do not claim that the regional emission patterns in poorly sampled continents like South America will be very accurate – there simply is not enough information available there, as the reviewer correctly points out. However, by having accurate emission information in well sampled regions and an additional strong constraint on the global emissions (which inversions with shorter backward simulations do not have), emissions in poorly sampled regions will also be corrected. Like global box model variants (e.g. AGAGE 8-box model) are able to retrieve emissions in several latitude bands, long FLEXPART simulations should also be able to give some information on the location of the emissions, even though the resolution will be very coarse.

Therefore, there should be enough information in the simulations that the inversion does not distribute the residual emissions (global minus well-constrained ones) randomly on the globe but still has enough skill to attribute them to the correct hemisphere, and hopefully also to the right continent at least. We discuss this in the text (see comment to L12, Final remark)

Similarly, care needs to be taken in the discussion of how different choices of baseline method should be used. There are examples where statistical baselines (perhaps including some baseline optimization) have provided consistent results to methods similar to GDB (e.g., Rigby et al., 2019; Brunner et al., 2017). Therefore, I think it is too broad to draw the conclusion that these methods need to be "abandoned" (Line 517). Rather, perhaps the case needs to be made that careful consideration should be given to the type of problem in which they are used.

We agree that our conclusion was perhaps too strong and have changed that in the text (see reply to the comment to L517).

Specific comments:

L6: suggest rewording: "… that purely statistical baseline methods CAN cause large systematic errors" (note "systematic", rather than "systematical")

➤ changed accordingly

L7: suggest removing "highly" before sensitive

➤ changed accordingly:

L8: In the final part of this sentence, I don't think it's quite fair to say "and that are consistent with recognized global total emissions". As discussed, I feel that agreement is primarily because the "recognised" emissions are used as the prior, and therefore these prior values are retrieved for integrated areas far from the measurement sites either because the particles haven't reached them yet (short simulations) or because the SRR is very low (even for long simulations). Of course, by design, and as demonstrated, the GDB method does a better job at accounting for the "missing" SRR in short simulations.

➤ We agree, that the good agreement for all simulation periods is partly due to the prior, especially for short backward simulation periods. In Fig. 14 we show the case of a strongly biased prior, where inversion results deviate strongly from the known global totals for short simulation periods but improve with longer periods. We think it is still important to mention that with the GDB method, global emissions stay close to the global prior, which is not the case for the other investigated methods.

and that are consistent with recognized global total emissions → and that show a better agreement with recognized global total emissions

L12: "Further, longer periods help to better constrain emissions in regions poorly covered by the global SF6 monitoring network (e.g., Africa, South America)". As discussed, I don't think this case has been made in this paper. I think this sentence should be removed.

> As mentioned before, we think the statement is actually true, but we weakened it and address your concerns in more detail in the discussion part.

Further, longer periods help to better constrain emissions in regions poorly covered by the global SF$_6$ monitoring network (e.g., Africa, South America) → Further, longer periods might help to better constrain emissions in regions poorly covered by the global SF$_6$ monitoring network.

> We added L488:
> > Final remark

In this study, we show many advantages for using relatively long backward simulation periods for the inversion. Nevertheless, the improvement of regional emission patterns is still limited by the observation network. A lack of observations in one region cannot simply be compensated by extending the simulations for stations in other regions to very long periods. For backward simulation times of 20-50 days, the emission sensitivity is distributed over large areas but usually still concentrated within broad latitude bands. The additional information to be gained from such long simulation times, on top of the information provided by the shorter simulation times, can probably best be compared with the inversions done with a multi-box model such as the AGAGE 8-box model (e.g. Rigby et al. ;2013) that is capable of determining the emissions in broad latitude bands. Consequently, if the emissions in certain regions with a dense observation network are already well constrained by shorter simulation periods, the residual emission will be attributed correctly as an emission total to all other regions of the same latitude band with a poor station coverage. The effective resolution of the obtained emissions in such data-poor regions may be very coarse but the result might still be informative. Furthermore, the emission sensitivity for the 20-50 day backward period is still not uniformly distributed over a latitude band and thus provides some limited regional information. Perhaps supported with a limited number of strategically located flask measurements, inversions with long backward simulation times could provide coarse but robust information on emissions in poorly sampled regions. Independently, the growing correlation between modeled and observed mixing ratios with increasing backward simulation length (Table 2; averaged over all stations) also shows that longer backward simulations hold additional information, even though the information gain decreases with every day added to the simulation length and probably becomes marginal for very long backward simulation times. However, we propose to make use of this additional information and apply longer periods whenever possible to make the best use of the existing observation network.

L33: "frequency", rather than "frequent"

> changed accordingly:

L37 – 39: Care needs to be taken with the periods ascribed to each study here. E.g., Brunner et al. (2017) uses LPDM runs from 5 to 19 days in length (see, Table 1 in that paper), not just 5 days; Rigby et al. (2019) has LPDM runs of 10 or 30 days, not just 10 days.

> changed accordingly:

…. that they are often run backward in time only for a few days ….

Brunner et al., 2017 → Vollmer et al., 2009

Rigby et al., 2019 → Thompson et al., 2017

L60 – 68: Other approaches should be cited here. E.g., Hu et al. (2019) compared a GDB-type approach to statistical methods, Lunt et al. (2016) uses a GDB method that uses mole fraction "curtains" around a regional domain (e.g., the termination points are tracked in space, rather than time), Rigby et al. (2011) and Ganshin et al. (2012) developed a nested Eulerian/Lagrangian approaches. It should be emphasised that in many of these papers, the "baselines" are adjusted as part of the inversion (so consider adding to the list on L155).

➤ changed accordingly:

Apart from using observations at each individual station to maintain a baseline, Rödenbeck et al. (2009) suggested a general "nesting" scheme, where a regional transport model – either a Eulerian or Lagrangian model – is embedded into a global model providing information from outside the spatio-temporal inversion domain. Such a global distribution based (GDB) approach was used by e.g. Trusilova et al. (2010) and Monteil and Scholze (2021) for carbon dioxide, and similar by Thompson and Stohl (2014) for methane. Whereas Rödenbeck et al. (2009) coupled the LPDM back-trajectories with the global model in the space domain, Thompson and Stohl (2014) did the coupling at the time boundary. →

Apart from using observations at each individual station to maintain a baseline, Rödenbeck et al. (2009) suggested a general "nesting" scheme, where a regional transport model – either a Eulerian or Lagrangian model – is embedded into a global model providing information from outside the spatio-temporal inversion domain. Such a global distribution based (GDB) approach was used by many authors: Trusilova et al. (2010) and Monteil and Scholze (2021) used Rödenbeck's approach to estimate $CO_2$ emissions. Similarly, Rigby et al. (2011) and Ganshin et al. (2012) developed approaches to nest a Lagrangian into a Eulerian model and tested it for $SF_6$ and $CO_2$, respectively. Estimating $CO_2$ baseline mole fractions for inverse modeling, Hu et al. (2019) applied two GDB approaches and a statistical method, where a subset of observations with minimal sensitivity was selected to correct a GDB baseline. Lunt et al. (2016) and Thompson and Stohl (2014) applied GDB approaches to model $CH_4$ While Thompson and Stohl (2014) coupled the LPDM back-trajectories with the global model at the end of the trajectories (which are terminated after a defined time), Lunt et al. (2016) used the exit location of the particles leaving the inversion domain for the coupling.

Added to list at L155: Rigby et al. (2011)

Figure 2: How have SRR due to flasks and high-frequency sites been combined here? Since they have a very different frequency, has there been any effort to "weight" their influence? If not, it may be worth adding this as a caveat. I.e., even though the flask footprint might look quite substantial, it corresponds to very few data points (and therefore relative influence on an inversion).

Indeed, for the figure the samples were not weighted by the number of observations at individual sites. However, we changed the figure in the revised manuscript to account for the variable sampling frequency at different sites, by weighting the sensitivities with the respective observation number.

[Figure]

Caption Figure 2: c) shows the SRR for the case of using surface flask measurement sites in addition to in situ measurements and for a 50 day simulation period. → c) shows the increase in the annual averaged SRR due to the use of flask measurements in addition to continuous measurements for the case of a 50-day backward simulation period.

L181: When also using surface flask measurements (Fig. 2c) in addition to in situ measurements for the case of a 50 day backward simulation period, the emission sensitivity is substantially higher almost everywhere and more smoothly distributed over the globe. However, regions of low sensitivity remain in the Tropics and in the Southern Hemisphere. → Figure 2c shows the increase in the annual averaged SRR due to the use of flask measurements in addition to continuous measurements in the case of 50-day simulations. One can see substantial increases in the vicinity of the measurement sites, that quickly decline with distance to the sites. Further SRR values increase in large parts of the Southern Hemisphere, however, the increases over southern continental areas are relatively low, as most flask measurements are not well located for inversion purposes.

L241: The use of "eliminate errors" and "any bias" is too strong. You can't eliminate errors or bias.

➢ changed accordingly

… on the ability to eliminate errors, and especially any bias of the global 3D mixing ratio fields … → … on the ability to minimize errors, and especially bias of the global 3D mixing ratio fields …

L245: Second sentence of this paragraph needs rewording, as it's not clear what the "It" refers to.

➢ changed accordingly

… provide a detailed description of FLEXPART CTM and evaluate it for the example of methane.
… provide a detailed description of FLEXPART CTM and evaluate this model for the example of CH4.

L263: Suggest "A priori emissions", rather than "information"

➢ changed accordingly:

L267 – 268: as noted in the preamble, it should be noted here that this choice of prior introduces some "circularity" into some of the results.

➢ we added:

Note at this point that the a priori emissions as constructed agree with recognized global emissions, which should be kept in mind when the global total is used as a reference value in the discussion.

Section 3.1: I think it needs to be mentioned here that both of these stations are somewhat complex in terms of baseline estimation, in that they periodically intercept air from the southern hemisphere. As noted, at Gosan, the summer months are characterised by southern hemisphere baselines. Therefore, I think these stations are likely to be among the most challenging in the world for statistical baseline estimation. I think the investigation would be improved if a station with less complex baseline were also added. For example, are similar results obtained if Mace Head is used?

➢ We chose these stations, as they are ideal to discuss the differences between the three investigated baseline methods. We agree however that Gosan and Barbados are challenging and that statistical methods might work better at less complex ones. We will discuss this in the text and refer to figures in the supplement, where we show the timeseries for all stations. We added:

L281: Both, Gosan and Ragged Point periodically intercept air from the southern hemisphere and therefore have a rather complex baseline.

L363: On the other hand, statistical baseline methods might work better at observation stations, where the baseline termination is less complex. At Mace Head (Fig. S18) for

instance, both REBS and Stohl's method lead to a very high correlation between modeled and observed mixing ratios for the case of a 50-day backward simulation ($r^2$=0.87). Nevertheless, for the REBS method, the discussed growing negative bias with longer simulation periods can be observed.

L287: "… as a result the baseline should become lower and smoother in order to leave a priori mixing ratios unchanged." I don't think this statement applies to REBS (it's presented as applying to all methods)

➢ The statement was meant in the sense, that this would be desirable (for a good baseline method)

Ideally, the choice of the backward simulation period should have no systematic effect on the calculated a priori mixing ratios. By increasing the backward simulation time, and therefore enlarging the temporal domain, more direct emission contributions are included. All these direct emission contributions should be removed from the baseline and as a result the baseline should become lower and smoother in order to leave a priori mixing ratios unchanged. Furthermore, one can assume that a correctly working baseline method leads to a proper agreement between *a priori* mixing ratios and observations. This agreement is investigated here for the three methods with → Ideally, the choice of the backward simulation period should have no systematic effect on the calculated a priori mixing ratios. By increasing the backward simulation time, and therefore enlarging the temporal domain, additional emission contributions are included in the optimization. Per definition, these contributions are not part of the baseline and should ideally be removed from it. As a result, the baseline should become lower and smoother when the simulation period is increased. We investigate the agreement between modeled and observed mixing ratios for the three methods with ….

L289: "… leads to a proper agreement between a priori mixing ratios and observations". I'm not sure what this means (the use of the term "proper").

➢ changed accordingly, see L287

L295: suggest removing "… when direct emission contributions get more impact", as it's not clear what this means, and seems to be unnecessary.

➢ changed accordingly

L307: I think it's too strong to say that Ragged Point is "uninfluenced" by polluted air masses. Pollution events are observed. They just tend to be very small (and/or well captured by short LPDM runs, since any sources are likely to be very local). Also note that L314 seems to conflict with this line, because it references an increasing direct emission contribution.

➢ changed accordingly:

Since Ragged Point is uninfluenced by regional emissions, no significant measurement peaks need to be excluded → Since regional pollution events captured at Ragged Point tend to be very small, no significant measurement peaks need to be excluded

L332: "… can only reproduce a few pollution events at Gosan…". Is this demonstrated in the 0-day simulation? If so, say so explicitly.

➢ Yes, we changed accordingly:

… it can only reproduce a few pollution events at Gosan, underestimates the highest and overestimates the lowest measured $SF_6$ mixing ratios (Fig. 6a). → … it can only reproduce a few pollution events at Gosan, underestimates the highest and overestimates the lowest measured $SF_6$ mixing ratios, as demonstrated in the 0-day case (Fig. 6a).

L334: "…provides, in principle, infinite resolution". I suggest this should be reworded, as infinite resolution isn't possible (for computational reasons and the resolution of the meteorology).

➢ changed accordingly:

…provides, in principle, infinite resolution → … provides much higher resolution

L340: Suggest rewording to: "… reproduces the measured mixing ratios well. However, it generates more variability than observed at this station"

➢ changed accordingly

L352: "Neither the REBS nor Stohl's method could correctly reproduce these negative SF6 excursions". As noted on the overarching comment for this section, this isn't surprising, as these methods aren't really designed for such complex baselines.

➢ Yes, it is not surprising that the statistical baselines do not work well for these stations. However, studies have applied these statistical methods to Gosan and Barbados (e.g. Fang et al. 2012, Stohl et al. 2009, 2010, Vollmer et al., 2017).

L354: "… it reproduces measurements insufficiently…". Not sure what this means. Do you mean the simulation can be biased?

➢ changed accordingly:

Despite of all advantages of the GDB method, it reproduces measurements insufficiently if the modeled global mixing ratio fields are biased. → Despite of all advantages of the GDB method, it doesn't work well if the modeled global mixing ratio fields are biased.

L367: Remove comma after "surprising"

➢ changed accordingly

L371: Remove "This is quite a substantial improvement." This is a subjective judgement. Leave this up to the reader to interpret.

➢ changed accordingly

Figure 9: Show posterior uncertainties.

➢ changed accordingly:

[Figure]

**Caption Fig.9**: National SF$_6$ emissions for selected countries, based on 20 day LPDM backward calculations with different choices of the baseline method. Uncertainties represent a 1σ range.

L384 – 392: There's an implicit assumption in this paragraph that the GBD method is "just right", and that the other methods are too high or too low. At this stage, we can't be sure which one is right or which is wrong. We can only compare one method with another.

➢ Yes, we agree, as the real emissions are unknown, we cannot be sure, what is right or wrong. However, when considering the increments together with the biases in Tab.2, we think there is good reason to make this assumption. We therefore discuss your comment and make this clear in the text!

When using the REBS method (Fig. 8b), the inversion produces negative emission increments in almost all areas of the globe, indicating that calculated baselines are too high overall. This is consistent with the assumption that the method overestimates the baseline at individual stations by wrongly classifying observations as baseline observations that are actually influenced by emissions within the backward calculation period. In contrast, the inversion algorithm produces positive increments almost everywhere around the globe when applying Stohl's method (Fig. 8c), suggesting that the method systematically underestimates the baseline (not only at background stations) which generally leads to a priori emissions that are too high. In case of the GDB method (Fig. 8d) negative and positive increments are more balanced, showing no sign of a systematical under- or overestimation of the baseline. Large positive increments can be seen in East Asian regions and parts of Europe, whereas the inversion tends to produce slightly negative increments in the Southern Hemisphere. →

When using the REBS method (Fig. 8b), the inversion produces negative emission increments in almost all areas of the globe. As the real emissions are unknown, this is not necessarily an unrealistic result. However, when considering these mostly negative increments together with the discussed positive bias for REBS baselines in Table 2 (especially for longer backwards simulation periods), there is reason to assume that the REBS method overestimates baselines and consequently underestimates the *a posteriori* emissions overall. In contrast, the inversion algorithm produces positive increments almost everywhere around the globe when applying Stohl's method (Fig. 8c). Again, considering this together with the discussed negative biases in Tab. 2, this might indicate an underestimation of the baselines and an overestimation of the *a posteriori* emissions overall. In case of the GDB method (Fig. 8d) negative and positive increments are more balanced. Overall the patterns are more similar to the ones of the REBS method, except in East Asia, where they rather resemble the patterns of Stohl's method. Large positive increments can be seen in East Asian regions and parts of Europe, whereas the inversion tends to produce slightly negative increments in the Southern Hemisphere.

L396: Suggest re-wording: "…cases and therefore the baseline choice has little impact."

➢ changed accordingly

L399: "revealing systematic problems in the first two methods". Again, now do you know that GDB is right and the others suffer from systematic problems?

➢ changed accordingly:

In almost all cases the REBS method leads to smaller and Stohl's method to larger national emissions than the GDB method, again revealing systematic problems in the first two methods. Due to the large emissions in China these problems become especially apparent there →

In almost all cases the REBS method leads to smaller and Stohl's method to larger national emissions than the GDB method. Due to the large emissions in China the differences in *a posteriori* emissions become especially apparent there ….

L404 – 405: I suggest noting again that this introduces some element of circularity (or at least note that this adds some nuance into the interpretation of the results)

➢ changed accordingly:

Notice that this is the same value used to calculate the a priori emissions, so the line represents also the global a priori emissions, which should be kept in mind for the interpretation of the results.

L424: This wording is too strong: "ability to ensure a flawless transition between the forward", as it's not possible to have a "flawless" simulation. But note also my suggestion in the preamble as to how one might interpret this result in terms of the low sensitivity of much of the world to the observations.

➢ changed accordingly:

Considering the inversion results based on the GDB method, global emissions are in good agreement with the box model result for all tested backward simulation periods. Furthermore, global emissions stay almost unchanged for different backward simulation periods, demonstrating again the method's ability to ensure a flawless transition between the forward (Flexpart CTM) and backward calculation. →

Considering the inversion results based on the GDB method, global emissions are in good agreement with the box model result for all tested backward simulation periods, as the global *a posteriori* emissions stay close to the global a priori value. Furthermore, these global emissions stay almost unchanged for different backward simulation periods, demonstrating the method's ability to adjust the baseline according to the sampled emissions of different simulation periods.

L433: I don't think this is truly "exponential"

➢ changed accordingly (see comment to L433 – 434)

L433 – 434: "For these poorly-monitored countries only backward simulations beyond the usual 5-10 days used in most studies provide information for the inversion". I disagree that 5 – 10 days is "usual". But as I said at the beginning, I don't think it's correct to imply that we can gain valuable new information from this length of simulation.

➢ We weakened the statement accordingly:

For these poorly-monitored countries only backward simulations beyond the usual 5-10 days used in most studies provide information for the inversion. For these countries, the SRR increase with time flattens to a linear increase only for very long transport times, even beyond the 50 days used in this study. →

For countries poorly covered by the monitoring network, however, the SRR is close to zero for the first 5 to 15 backward days and only longer backward simulations might provide information for the inversion (see Fig. 11b). For these countries, the SRR increase with time flattens to a linear increase only for very long transport times, even beyond the 50 days used in this study.

Figure 13 caption: Some indication of the significance of this difference would be useful.

➢ We added a plot to Fig. 13 showing the additional error reduction due to the use of the flask measurements (we also changed the color bar to better distinguish it from Fig. 12,)

L450: "illustrating the great value of this additional information". We need to see the uncertainties before we can decide if this demonstrates great value. Are any of these changes significant?

➢ Yes, we agree. We rewrote this section and discuss the changes together with the additional error reduction. Also, we added additional text and figure to discuss the impact of the additional flask data on the national and global emission estimates.

[Figure]

Caption Fig 13: a) Relative change in *a posteriori* emissions and b) the additional error reduction when using flask measurements in addition to in situ measurements for the 50-day simulation. The locations of the flask measurements are marked with black dots.

Figure 13a) shows the relative change in a posteriori emissions and Fig. 13b) the additional relative error reduction when using flask measurements additionally to the in situ measurements for the 50-day backward simulation. One can see substantial differences in the USA, Eastern Europe, South Africa, East Asia and the Near East, where also an additional error reduction occurs. While this additional error reduction can be relatively large (up to 73 %) for grid cells in the vicinity of the measurement sites, it quickly decreases down to a few percent with larger distance to the measurements. Consequently, flask measurements show only small influence on the total global emission estimate (< 1%), but can have a large impact on calculated national emissions of specific countries (Fig. A4). For countries in the Near East the additional use of flask measurements changes national emission estimates by 40 to 100%. South African and American emissions are modified by around 10%.

[Figure]

Caption Fig A4: relative change in national *a posteriori* emissions of selected countries, when flask measurements are used in addition to continuous measurements in the case of 50-day simulations.

L467: "However, it is clear that a substantial bias remains even with a backward simulation period of 50 days. It seems likely that an extension of the backward simulation period beyond 50 days would further reduce the bias." As I said, I don't think this is true. I think this likely comes from low sensitivity to emissions far from the measurement stations. Otherwise, the implication would be that we could overcome any bias in the prior using just one station and a very long simulation

➢ As mentioned above, we think this statement is actually true. In principle, even with one station and very long simulation periods, global emissions should be obtainable (similar to a box model). In our case, the total emission estimates should improve with increasing backward simulation time, however with varying spatial resolution of the emission patterns strongly depending on the observations network. We discuss this in the text (see comment to L12, Final Remark).

L494: remove "entirely"

➢ changed accordingly

L502: "… is superior and leads to a posteriori emissions that are far less sensitive to the LPDM backward calculation length and that are consistent with global total emissions". I think you can only say: "… leads to a posteriori emissions that are less sensitive to LPDM backward calculation lengths than the other baseline estimation methods tested here"

➢ changed accordingly:

is superior and leads to a posteriori emissions that are far less sensitive to the LPDM backward calculation length and that are consistent with global total emissions → leads to a posteriori emissions that are less sensitive to LPDM backward calculation lengths and stay close to the global total emission value.

L512: "… improves the observational constraint on SF6 emissions substantially". Again, I think we need to know how significant this result is. (Not enough to just demonstrate that the mean has changed in some regions).

➤ Yes, we agree and changed the sentence accordingly:

The additional use of flask measurements improves the observational constraint on $SF_6$ emissions substantially. → The additional use of flask measurements has the potential to improve the observational constraint on $SF_6$ emissions, especially close to the measurement sites.

L517: "Following these results, we strongly recommend to abandon the use of baseline methods based purely on the observations of individual sites, for inverse modelling". I think this statement is too strong. Clearly, studies have shown statistical methods to be consistent with GDB methods for some regions/approaches. My feeling is that you need to be very careful when and where you use them.

➤ Yes, we agree and changed the sentence accordingly:

Following these results, we strongly recommend to abandon the use of baseline methods based purely on the observations of individual sites, for inverse modelling → Following these results, we advise against the use of baseline methods that are purely based on the observations of individual sites. At least great care needs to be taken that problems such as demonstrated in this paper do not occur.

L519: again, I'm not sure 5-10 days is "usual".

➤ changed accordingly:

We also recommend to employ longer LPDM backward simulation periods, beyond the usual 5-10 days, as this leads to improvements in overall model performance, helps to constrain emissions in regions poorly covered by the monitoring network, and produces more robust global emission estimates. → We recommend also to employ longer LPDM backward simulation periods, beyond 5-10 days, as this can lead to improvements in overall model performance, can produces more robust global emission estimates and might help to constrain emissions in regions poorly covered by the monitoring network.

L520: "When consistency between regional and global emission estimates is important, even longer backward simulation periods than 50 days may be useful." Again, I don't think you can derive global emissions with very long simulation lengths in the real world. There are other factors that get in the way (low sensitivity, accumulation of errors).

➤ As mentioned, we think this statement is actually true. Globally, the sensitivity will grow for longer simulation periods and we think that the growing error of individual trajectories will become less and less important (due to the statistical approach of FLEXPART looking at average residence times rather than the individual trajectories). When run long enough (years), FLEXPART produces a well-mixed state of particles (where particle densities are proportional to air density). This is then equivalent to a global box model, and these models have been used for a long time to estimate global emissions. We discuss this in the text (see comment to L12, Final remark).

**References**

Brunner, D., Arnold, T., Henne, S., Manning, A., Thompson, R. L., Maione, M., O'Doherty, S., and Reimann, S.: Comparison of four inverse modelling systems applied to the estimation of HFC-125, HFC-134a, and $SF_6$ emissions over Europe, Atmos. Chem. Phys., 17, 10651–10674, https://doi.org/10.5194/acp-17-10651-2017, 2017.

Ganshin, A., Oda, T., Saito, M., Maksyutov, S., Valsala, V., Andres, R. J., Fisher, R. E., Lowry, D., Lukyanov, A., Matsueda, H., Nisbet, E. G., Rigby, M., Sawa, Y., Toumi, R., Tsuboi, K., Varlagin, A., and Zhuravlev, R.: A global coupled Eulerian-Lagrangian model and 1 × 1 km CO2 surface flux dataset for high-resolution atmospheric CO2 transport simulations, Geoscientific Model Development, 5, 231–243, https://doi.org/10.5194/gmd-5-231-2012, 2012.

Hu, L., Andrews, A. E., Thoning, K. W., Sweeney, C., Miller, J. B., Michalak, A. M., Dlugokencky, E., Tans, P. P., Shiga, Y. P., Mountain, M., Nehrkorn, T., Montzka, S. A., McKain, K., Kofler, J., Trudeau, M., Michel, S. E., Biraud, S. C., Fischer, M. L., Worthy, D. E. J., Vaughn, B. H., White, J. W. C., Yadav, V., Basu, S., and van der Velde, I. R.: Enhanced North American carbon uptake associated with El Niño, Sci. Adv., 5, eaaw0076, https://doi.org/10.1126/sciadv.aaw0076, 2019.

Lunt, M. F., Rigby, M., Ganesan, A. L., and Manning, A. J.: Estimation of trace gas fluxes with objectively determined basis functions using reversible-jump Markov chain Monte Carlo, Geoscientific Model Development, 9, 3213–3229, https://doi.org/10.5194/gmd-9-3213-2016, 2016.

Rigby, M., Manning, A. J., and Prinn, R. G.: Inversion of long-lived trace gas emissions using combined Eulerian and Lagrangian chemical transport models, Atmospheric Chemistry and Physics, 11, 9887–9898, https://doi.org/10.5194/acp-11-9887-2011, 2011.

Rigby, M., Park, S., Saito, T., Western, L. M., Redington, A. L., Fang, X., Henne, S., Manning, A. J., Prinn, R. G., Dutton, G. S., Fraser, P. J., Ganesan, A. L., Hall, B. D., Harth, C. M., Kim, J., Kim, K.-R., Krummel, P. B., Lee, T., Li, S., Liang, Q., Lunt, M. F., Montzka, S. A., Mühle, J., O'Doherty, S., Park, M.-K., Reimann, S., Salameh, P. K., Simmonds, P., Tunnicliffe, R. L., Weiss, R. F., Yokouchi, Y., and Young, D.: Increase in CFC-11 emissions from eastern China based on atmospheric observations, Nature, 569, 546–550, https://doi.org/10.1038/s41586-019-1193-4, 2019.

Thompson, R. L., Sasakawa, M., Machida, T., Aalto, T., Worthy, D., Lavric, J. V., Lund Myhre, C., and Stohl, A.: Methane fluxes in the high northern latitudes for 2005–2013 estimated using a Bayesian atmospheric inversion, Atmospheric Chemistry and Physics, 17, 3553–3572, https://doi.org/10.5194/acp-17-3553-2017, 2017

Vollmer, M., Zhou, L., Greally, B., Henne, S., Yao, B., Reimann, S., Stordal, F., Cunnold, D., Zhang, X., Maione, M., et al.: Emissions of ozone-depleting halocarbons from China, Geophysical Research Letters, 36, https://doi.org/10.1029/2009GL038659, 2009

---

## Author Comment (AC2)

**Author's response #2**

**General**

The manuscript " is an important contribution to the field of inverse modelling of synthetic gases. While I largely agree that the use of global concentration fields in combination with backward Lagrangian modelling is superior to the use of observation-derived baselines and should be employed wherever possible, I still feel that the present study is not sufficiently general to arrive at the strong conclusions and recommendations it makes. There certainly are situations and alternative descriptions for which observation-based approaches can yield unbiased emission estimates and, due to the lack of realistic three-dimensional representations of the target compound, present the only reasonable approach. Otherwise the presented study is of high quality, well presented and should be published as soon as the points below were addressed.

We would like to thank reviewer #2 for the very helpful and productive review of our manuscript.

In the response we use 4 different colors. The blue colored text is the general answer to the reviewer's comments. Additionally, we show how the text is changed in the manuscript: The original text is colored grey, removed text is colored red, and new text is colored green.

**Major comments**

Very general conclusions: The discussion and conclusions of the present study are not well balanced. They are generalizing the findings of this specific study in a way that seems overreaching. This becomes especially apparent in the conclusion section where very general recommendations are made. From the presented results it is only apparent that certain inversion setups do not work well. However, the current study is incomplete in the sense that only a relatively specific kind of baseline treatment was analyzed in a single global inversion system, while there are several other systems and approaches used by other research groups and it remains unclear if these other approaches suffer from the same problems. The next points give two alternatives to the solution suggested in the present study. These should at least be considered in the discussion of the manuscript and the conclusions should be amended accordingly.

We agree that some of our conclusions were very general, as the reviewer points out. While incorporating the suggestions, we took great care to adapt our statements, to not generalize our findings beyond their validity.

No consideration of other commonly used baseline methods: The paper focusses on two observation-base baselines methods that have been used in regional inversions of synthetic gases (REBS and Stohl), but it does not mention that other approaches exist (beside the suggested GDB method). Most notably, the method applied in the UK by the MetOffice and Uni Bristol groups seems worth mentioning (see for example Manning et al. (2021) and references therein). While their method (similar to Stohl's method) combines observations and model information, it does not assume that the baseline is a smooth curve, but it depends on the direction and height of the air entering a regional domain. In this way is it possible to describe temporally varying baselines that do not suffer from some of the problems described in the current paper. Admittedly, their method is supposed to be used for regional-scale, limited area inversions. However, it has been used before in a combined Eulerian/Lagrangian study on SF6 (Rigby et al., 2011) and is certainly worth mentioning.

Also, we appreciate that it was pointed out that the manuscript didn't mention other important baseline approaches. We rewrote parts of the introduction including the suggested methods (O'Doherty et al. 2001; Rigby et al., 2011; Manning et al., 2021) and others (Ganshin et al. 2012; Lunt et al. 2016; Hu et al. 2019).

No optimization of baseline: Most regional scale inversion studies that utilize some kind of baseline estimate (if purely observation-based, including transport information, or larger scale concentration fields) don't deny that there may be problems with these assumed/derived baselines. As a consequence, they optimize the baseline in some way in the inversion. Here, the authors chose to apply baselines without further optimization and come to the conclusion that simple baseline approaches should be avoided. However, it would be interesting to see how much an appropriately configured baseline optimization step could remedy some of the problems encountered for example for the REBS and Stohl baselines. Such a test would also be very valuable in the context of biased larger scale concentration fields (see L241, L354), such as used here from FLEXPART-CTM, but more general when derived from satellite data assimilation. I would encourage the authors to do an additional set of inversions where they optimize the REBS and/or Stohl baselines and add such results to Figure 10. In addition, similar tests could be done with the biased baseline fields as presented in Figure 14 c/d.

The optimization of the baseline is a purely statistical correction and may falsely compensate for errors elsewhere (e.g., in the emissions). Therefore, we think it is in any case desirable to obtain a baseline that is as accurate as possible prior to any optimization. For that reason, we abstained from optimizing the baseline in our original manuscript. However, we agree that the manuscript could be improved by also including such an optimization of the baseline in our investigation. We accept the proposal to do an additional set of inversions with optimized baselines, to see how much this could remedy some of the encountered problems.

L155:

In contrast to many other studies (e.g., Henne et al., 2016; Stohl et al., 2009; Thompson and Stohl, 2014) we do not use the option to optimize the baseline mixing ratios in the inversion. This gives us the opportunity to better analyze the differences between investigated baseline methods and to study their impacts on the a posteriori emissions in detail → In contrast to many other studies (e.g., Ganshin et al. (2012), Henne et al., 2016; Rigby et al. (2011), Stohl et al., 2009; Thompson and Stohl, 2014) we do not use the option to optimize the baseline mixing ratios in the inversion, except for sensitivity tests. In any case, it is desirable to obtain a baseline that is as accurate as possible prior to any optimization, which is a purely statistical correction that may falsely compensate for errors elsewhere (e.g., in the emissions). Waiving this option gives us further the opportunity to better analyze the differences between investigated baseline methods and to study their impacts on the *a posteriori* emissions more systematically. For the baseline optimization of the sensitivity tests, we use a temporal window of 28 days and a baseline uncertainty of 0.1 ppt. Increasing this value up to 0.2 ppt did not show any significant changes in the results. For general details on the baseline optimization see Thompson and Stohl, 2014.

L426: We further investigate whether the encountered biases can be reduced by optimizing the baseline in the inversion. Therefore, we repeated the inversion with exactly the same setup, except optimizing the REBS and Stohl's baseline as part of the inversion. Results are shown in Fig. A2. In case of the REBS method the baseline optimization has only little effect on the global total *a posteriori* emissions for backward simulation periods between 1 and 10 days and becomes noticeable only after 20 days. The greatest

improvements can be observed for the 50-day simulation, where the bias is almost halved. Still, for longer simulation periods the increasing improvements through the baseline optimization cannot compensate for the growing underestimation of the emissions and substantial biases remain. Optimizing Stohl's baseline shows great improvements, especially for longer simulation periods. These improvements increase systematically with growing backward simulation period and results get very close to the box model outcome for the 20- and 50-day simulation case.

[Figure]

Figure A2: Calculated SF$_6$ global emissions when baseline concentrations are optimized as part of the inversion. Grey bars represent the improvements obtained by the baseline optimization. Results are shown for the REBS and Stohl's method and for all five applied simulation periods between 1 and 50 days. The horizontal dashed line represents the reference value of the AGAGE 12-box model with shaded error bands.

**Minor comments**

L25: Not sure that Henne et al. (2016) is the most general reference for promoting inverse modeling methods. Articles like Nisbet and Weiss (2010), Weiss and Prinn (2011) or Leip et al. (2018) have a much broader claim on the subject. Just to mention a few.

➢ changed accordingly:

   (Henne et al. 2016) → (e.g. Leip et al. 2018; Weiss and Prinn 2011)

L32 and elsewhere: The term 'in-situ' seems to be used to distinguish continuous from flask sampling. In my understanding of the term, both are in-situ, even if the second is not analysed in the field. This is opposed to remote sensing observations. I would suggested to distinguish between continuous and flask sampling. The use of satellite observations for inverse modelling could be mentioned at this point of the manuscript as well.

➢ changed accordingly:

Most studies only use in situ observations for this purpose, however low frequent flask measurements can also be included (e.g. Villani et al., 2010) → Most studies only use continuous in situ observations for this purpose, however flask measurements with low sampling frequency can be included as well (e.g. Villani et al., 2010). For certain species, satellite measurements could also be used.

L34f: Already here, it would be important to distinguish between regional and global scale studies. Most (all) of the cited papers focus on regional scale for which longer integration times are not necessarily useful because particles will have left the regional domain for which emissions are determined!

We agree and added:

L38: 7 days (Koyama et al. 2011)

L39: The choices made …. → Koyama et al. (2011) and Stohl et al. (2009) are global inversion studies, while the other listed studies apply regional inversions. The choices of the used backward simulation period made …

L49f: I wonder why REBS was selected as a baseline method. It is not the official baseline method operationally applied within the AGAGE network and used for many regional and global inverse modelling studies. While the AGAGE method (also referred to as Georgia Tech method (O'Doherty et al. (2001)) is purely observation-based, methods using additional model information are commonly used within the AGAGE community as well (Manning et al. 2021). Adding some of this information to the introduction would be useful.

We admit that there are other observation-based (as well as partly model-based) baseline methods. We chose the REBS method because it has become popular with some authors. Furthermore, even though the statistical methods to derive these baselines may differ in details, they have similar properties. Thus, the REBS method should be representative of such methods. To mention also other methods, we have added (L48):

Such statistical methods have been operationally applied within observation networks, such as the Georgia Institute of Technology method (O'Doherty et al. 2001) used within the AGAGE community.

L57 We rewrote:

A baseline method introduced by Stohl et al. (2009), further termed as "Stohl's method", tries to avoid this baseline overestimation, by using model information to subtract prior simulated mixing ratios from pre-selected observations. Nevertheless, this pre-selection is subjective and prior simulated mixing ratios depend on a priori emission estimates.

Apart from using observations at each individual station to maintain a baseline, Rödenbeck et al. (2009) suggested a general "nesting" scheme, where a regional transport model – either a Eulerian or Lagrangian model – is embedded into a global model providing information from outside the spatio-temporal inversion domain. Such a global distribution based (GDB) approach was used by e.g. Trusilova et al. (2010) and Monteil and Scholze (2021) for carbon dioxide, and similar by Thompson and Stohl (2014) for methane. Whereas Rödenbeck et al. (2009) coupled the LPDM back-trajectories with the global

model in the space domain, Thompson and Stohl (2014) did the coupling at the time boundary.

→

In addition to the statistical selection some methods also use model information to improve the baseline. A method applied by the UK Met Office and commonly used within the AGAGE network (see e.g. Manning et al., 2021) identifies baseline measurements by analyzing the direction and height of air entering the regional inversion domain. A baseline method introduced by Stohl et al. (2009), further termed as "Stohl's method", uses model information to subtract prior simulated mixing ratios from pre-selected observations, in order to avoid an overestimation of the baseline. Nevertheless, this pre-selection is subjective and prior simulated mixing ratios depend on a priori emission estimates.

Apart from using observations at each individual station to maintain a baseline, Rödenbeck et al. (2009) suggested a general "nesting" scheme, where a regional transport model – either a Eulerian or Lagrangian model – is embedded into a global model providing information from outside the spatio-temporal inversion domain. Such a global distribution based (GDB) approach was used by many authors: Trusilova et al. (2010) and Monteil and Scholze (2021) used Rödenbeck's approach to estimate $CO_2$ emissions. Similarly, Rigby et al. (2011) and Ganshin et al. (2012) developed approaches to nest a Lagrangian into a Eulerian model and tested it for $SF_6$ and $CO_2$, respectively. Estimating $CO_2$ baseline mole fractions for inverse modeling, Hu et al. (2019) applied two GDB approaches and a statistical method, where a subset of observations with minimal sensitivity was selected to correct a GDB baseline. Lunt et al. (2016) and Thompson and Stohl (2014) applied GDB approaches to model $CH_4$. While Thompson and Stohl (2014) coupled the LPDM back-trajectories with the global model at the end of the trajectories (which are terminated after a defined time), Lunt et al. (2016) used the exit location of the particles leaving the inversion domain for the coupling.

L63-65: At this point it is not clear what you mean by space domain. I suppose all of the mentioned studies couple in space (domain boundaries) and time. Even Thompson and Stohl couple in time and space, since the sensitivity field is used, just like in the present study, right? I suggest to rephrase these two sentences and clarify the difference. Another study to mention here is that by Rigby et al. (2011), which followed a similar approach for SF6, but is not discussed for that reason so far.

➢   We agree and changed accordingly: See rewritten text L57.

L76f: Citing Rigby et al. (2011) would make sense here as well.

➢   changed accordingly:

Given that the LPDMs are usually run backward in time only for a few days, the inversions constrain the emissions only in regions where observation stations exist (Rigby et al., 2011).

L104/105: The observation treatment is a bit unclear. According to this sentence, observations in a four-hour window were selected (12:00 to 16:00) and then aggregated to 3-hourly intervals. Does this imply that two aggregates were formed (if observations were present). For example one for 12:00 to 15:00 and one for 15:00 to 18:00 if local time is UTC? Why the four-hour window instead of simply taking a single three-hour window aligned with the simulations?

Indeed, we chose a 4 hour window. We thought a 3 hour window would be too small, but as the reviewer points out, it would probably have been a more rational choice. However, we don't expect this choice to have a big influence on the inversion results.

L108: If I recall Stohl et al. (2009) correctly, they did not remove observations but assigned larger uncertainties to those where the mismatch was large. Doing so in an iterative step. Does the current approach, hence, differ from Stohl et al. (2009)?

Yes, that is correct. While Stohl et al. (2009) assigned larger uncertainties, we removed them. We make this difference clear in the text:

Additionally, we followed a method by Stohl et al. (2009) to remove observations that can not be brought into agreement with modeled mixing ratios by the inversion. → Additionally, we followed a method by Stohl et al. (2009) to identify observations that cannot be brought into agreement with modeled mixing ratios by the inversion, which we removed completely (in contrast to Stohl et al. (2009), who assigned larger uncertainties to these observations).

L112: Why was the year 2012 chosen for this study? Wouldn't there be more and more precise observations of SF6 for more recent years?

The year 2012 was actually the year for which we could collect the largest number of observations. For some stations (Cape Ochiishi, Izaña, Summit) data where only available up to a certain year.

L113: At this point it is not yet clear what the 're-analysis' is. This only becomes clear in section 2.3. Somehow introduce the re-analysis and motivate as well why two instead of one year of observations were needed (once more only clarified in 2.3)

For the re-analysis of $SF_6$ (see section 2.5) we used all the available 2011 and 2012 in situ measurements from the sites listed in Table 1 →

In order to generate global $SF_6$ mixing ratio fields required by the GDB method, we performed a two-year $SF_6$ re-analysis (for more details see section 2.5), for which we used all the available 2011 and 2012 continuous measurements from the sites listed in Table 1.

L143ff: Please mention number of grid cells in inversion grid. Later on (section 2.6), a temporal correlation is mentioned. Please clarify if x varies with time and if so what is the temporal resolution

We use a spatial emission grid (Fig. A1) with a varying grid size ranging from 1°x1° to 16°x16°. → We use a spatial emission grid (Fig. A1) with 6219 grid cells of varying size ranging from 1°x1° to 16°x16°.

We thank the reviewer a lot for raising this point. x does not vary with time – there was actually a mistake in the text in section 2.6

L149: We added: **x** is assumed to not vary with time.

L271: Spatial and temporal correlation between uncertainties are considered by using an exponential decay model with a scale length of 250 km and 30 days. The error covariance matrix B is calculated as the Kronecker product of the spatial and temporal covariance matrices. → Spatial correlation between uncertainties are considered by using an exponential decay model with a scale length of 250 km.

Fig2: Does the presented source receptor relationship take the variable sampling frequency at different sites into account? To me it looks as if the flask sampling sites have a very similar source receptor relationship as the continuous sites, whereas in reality they should have total sensitivities at least one order of magnitude smaller (one weekly sampling vs two 3-hour aggregates per day).

Indeed, for the figure the samples were not weighted by the number of observations at individual sites. However, we changed the figure in the revised manuscript to account for the variable sampling frequency at different sites, by weighting the sensitivities with the respective observation number.

[Figure]

Caption Figure 2: c) shows the SRR for the case of using surface flask measurement sites in addition to in situ measurements and for a 50 day simulation period. → c) shows the

increase in the annual averaged SRR due to the use of flask measurements in addition to continuous measurements for the case of a 50-day backward simulation period.

L181: When also using surface flask measurements (Fig. 2c) in addition to in situ measurements for the case of a 50 day backward simulation period, the emission sensitivity is substantially higher almost everywhere and more smoothly distributed over the globe. However, regions of low sensitivity remain in the Tropics and in the Southern Hemisphere. → Fig. 2c shows the increase in the annual averaged SRR due to the use of flask measurements in addition to continuous measurements in the case of 50-day simulations. One can see substantial increases in the vicinity of the measurement sites, that quickly decline with distance to the sites. Further SRR values increase in large parts of the Southern Hemisphere, however, the increases over southern continental areas are relatively low, as most flask measurements are not well located for inversion purposes.

L245: If I understand correctly, the same observations are first nudged into FLEXPART-CTM and then used for the global inversion step. While, this seems to be great to remove any biases in baseline concentrations it also means that baseline and inversion are not independent, which may require additional considerations for the Bayesian inference.

A potentially cleaner implementation of the GDB method would be to strictly avoid the double use of observations, and thereby prevent what is often considered to be an inverse crime (Colton and Kress, 1992). However, this is challenging due to the limited number of $SF_6$ observations. Generally, we assume that the dependence of the baseline on the individual measurements is rather small, as baseline concentrations are calculated by averaging concentrations over many grid cells, especially for longer backward simulation periods, and they are influenced by many observations (and the model) simultaneously. For instance, with backward simulation times >30 days, monthly inversions should actually become independent from the observations defining the baseline. In this case, the observations are taken from the previous month and are not used for constraining the emissions during that particular month. This is another argument for using rather long backward simulation times. Strictly, this would be true only if no temporal correlation of the emissions is applied. However, the influence of the double-use would decrease substantially with such a setup. In contrast, the REBS method and Stohl's method use observations directly to maintain the baseline. We also want to emphasize that we see no artifacts associated with this double use and therefore assume the influence to be weak.

L247: What is the rational for using 12 million model particles? Later on, it is mentioned that this may limit the quality of the derived concentration fields. Would a doubling of the particle number have helped improving the concentration fields?

The choice was a compromise between accuracy and computational time. We doubled the number of particles used by Groot Zwaaftink et al. (2018), who simulated methane concentration fields. But yes, we would expect that a larger number of particles still improves the concentration fields. However, for long backward simulation times, for which the baseline is constructed based on a large number of grid cells, the difference would be minimal.

L250: Is a single year spin-up sufficiently long to get the vertical profiles into equilibrium?

We agree that a single year might have been not sufficiently long and that longer times might improve the global fields. However, since undiluted transport from higher altitudes is rare, the impact on the inversion is thought to be small.

L253: What is the temporal resolution of the output? Was it used in this resolution for the coupling to the backward simulations?

Mixing ratio fields are saved on a 3°x2° output grid and extrapolated to the same grid as the termination sensitivity fields → Mixing ratio fields are saved daily on a 3°x2° output grid and coupled to the backward simulations.

Sec 3.1: Although, the situation of the two stations is described, it is not spelled out why these two sites were selected. I assume to show one polluted vs one clean site. Findings for these sites may therefore be extreme. Please discuss this when introducing the sites and please add that Ragged Point, as an equatorial site, is intermittently impacted by southern and northern hemisphere air, which makes baseline estimation a challenging task (as seen later).

We changed that accordingly (notice also that the results for all stations can be found in the supplement):

L281: Both, Gosan and Ragged Point periodically intercept air from the southern hemisphere and therefore have a rather complex baseline.

L363: On the other hand, statistical baseline methods might work better at observation stations, where the baseline determination is less complex. At Mace Head (Fig. S18) for instance, both REBS and Stohl's method lead to a very high correlation between modeled and observed mixing ratios for the case of a 50-day backward simulations ($r^2$=0.87). Nevertheless, for the REBS method, the discussed growing negative bias with longer simulation periods can be observed.

L281f: One has to read until the end of the paragraph to get the link to the figures. Would be good to have this from the beginning of the description.

Agreed, we added the figure references here:

Baseline mixing ratios are plotted together with respective observations and a priori mixing ratios for different LPDM backward simulation periods ranging from 1 to 50 days (Fig 4-7).

L286f: The sentence is a bit hard to grasp. Consider rephrasing.

Changed accordingly:

Ideally, the choice of the backward simulation period should have no systematic effect on the calculated a priori mixing ratios. By increasing the backward simulation time, and therefore enlarging the temporal domain, more direct emission contributions are included. All these direct emission contributions should be removed from the baseline and as a result the baseline should become lower and smoother in order to leave a priori mixing ratios unchanged. Furthermore, one can assume that a correctly working baseline method leads to a proper agreement between *a priori* mixing ratios and observations. This agreement is investigated here for the three methods with → Ideally, the choice of the backward simulation period should have no systematic effect on the calculated a priori mixing ratios. By increasing the backward simulation time, and therefore enlarging the temporal domain,

additional emission contributions are included in the optimization. Per definition, these contributions are not part of the baseline and should ideally be removed from it. As a result, the baseline should become lower and smoother when the simulation period is increased. We investigate the agreement between modeled and observed mixing ratios for the three methods with ….

L299f: Agreed: but the remedy is to choose a backward integration time that fits the definition of the baseline. The backward integration time is usually chosen such that a released tracer would become well mixed within a latitude band with this period and therefore unobservable as such. This is usually assumed to be around two weeks. 50 days certainly is a bit long and, hence, some of what is background from the observational point of view gets mixed in into the 'recent' signal. It would be interesting to see the concentration increases between day 20 and 50. How variable are they? From Fig 7e/f this seems to be a rather constant contribution. This should be discussed along with the motivation for using smooth baselines.

We are not exactly sure what the reviewer means here. The concentration increases between 20 and 50 days are rather constant at most sites but not totally smooth, depending on site location. (See also the corresponding comment in L346) The REBS method certainly would work best with rather short backward simulation times but it is not clear what the "optimum" time would be.

L314f: But then again the absolute bias is larger than with the REBS method. So it's kind of difficult to say which method is superior here? There is another obvious problem with Stohl's approach for Ragged Point. The lowest concentrations are obviously (as shown later) due to southern hemispheric influences. Stohl's baseline, hence, is more representative for southern hemispheric conditions. However, these do not necessarily dominate at the site. Ragged Point is certainly a fine example where both methods are predestined to fail, because there is no such thing as a smooth baseline for this site due to the large inter-hemispheric concentration gradient and the intermittent hemispheric influence.

Yes, we agree:

In contrast, due to its 25th percentile pre-selection of observations, Stohl's method shifts the baseline curve towards the lower observations. For low direct emission contributions (Fig. 5a/b), a priori mixing ratios thus underestimate the observations. → In contrast, due to its 25th percentile pre-selection of observations, Stohl's method shifts the baseline curve towards the lowest observations. In the case of Ragged Point, these lowest observations come from southern hemispheric air masses. Hence, Stohl's baseline is more representative for southern hemispheric conditions, which do not necessarily dominate at that site. Consequently, a priori mixing ratios underestimate the observations for low direct emission contributions (Fig. 5a/b).

However, the rather ad hoc 25th percentile pre-selection of data for the baseline is obviously not justified for a background station with few pollution episodes and southern hemispheric air interceptions, leading to a systematic underestimation of modeled a priori mixing ratios, irrespective of the length of the backward simulation.

L320: When showing the comparison of the observations to FLEXPART-CTM output (0 day backward), it would be good to mention that one would expect a close agreement, since these observations were used for nudging in FLEXPART-CTM. So no big surprise that they fit so well in the case of Ragged Point.

We added: This good agreement is however expected, since these observations were used for the nudging in the FLEXPART CTM model.

L329/330: Somehow seems to contradict the later conclusion that long integration times are important.

From our point of view, this is not really a contradiction, although we see, what the reviewer means. On the one hand, we expect from a good baseline method that it accounts for different direct emission contributions from different simulation periods – hence, there should not be a really large systematic change in the a priori mixing ratio, for different simulation periods. On the other hand, we argue for longer simulation periods, as we see that overall the agreement between a priori modeled and observed mixing ratios improves with longer simulations (which, admittedly, is only slightly the case for Gosan). We add the word "systematic" to make this clear. Nevertheless, even if a priori mixing ratios would stay totally the same for longer backward simulations, there could still be an improvement in the posterior emissions, as more direct emission contributions can be optimized.

As a result a priori mixing ratios in Fig. 6 show no large systematic changes with increasing simulation period between 5 and 50 days

L356f: But the local FLEXPART-CTM cell should not have an impact for longer integration times. Could there be a bias between the observational data that was used for nudging over North America and the observations at Mace Head. These are two different networks, right? Then there is the possibility that the nudging over North America is not sufficiently strong to remove any bias introduced by North American emissions. Have you checked the CTM performance at the nudging locations? The inversion corrects East Coast emission down (Fig 8). So maybe they drive a baseline shift in the prior.

Yes, we agree:

Removed: At Mace Head, this could be explained to some extent by the close proximity of the station to the 3°x2° grid cell border resulting in the possibility that FLEXPART CTM attributes strong point emission contributions to the (relatively large) grid cells, that would be outside of the respective area of influence in reality. Other

We don't think that there is a bias between the observational data that was used for nudging over North America and the observations at Mace Head. The overestimation of the observations can also be seen for runs without nudging. Yes, we think that the nudging over North America is not strong enough to remove the bias introduced by North American emissions or biases in upper heights. A comparison to flight measurements (ones that were used for nudging, but also independent ones) shows that the model slightly overestimates concentrations in upper heights at higher latitudes.

L365f: In the presented case the GDB method is also not independent form the observations, since these were used for nudging. So it is not surprising that there is only a small bias!

Yes, that is right. Still we think that with the presented GDB method the dependency on the observations is still smaller, since they are not directly used to calculate the baseline, but only to improve the concentration fields. Consider for instance the lowest baseline concentrations at Ragged Point which depend mainly on the concentration in the Southern Hemisphere, (where back trajectories terminate). The dependency on the observations (used for nudging at that station) will be very small here. Generally, the dependency on

observations decreases with longer simulation period, as already discussed. Hence, we find it even more remarkable that the agreement between modeled and observed concentrations increases with longer simulation periods.

One should keep in mind that the REBS and Stohl's method are based on the observations themselves and thus the observed and modeled a priori mixing ratios are not independent. → One should keep in mind that the REBS and Stohl's method are directly based on the observations themselves and thus the dependency between observed and modeled a priori mixing ratios is likely higher than in the case of the GDB method, where observations are rather used to improve the mixing ratio fields.

L367: I think the opposite is true. It is really surprising that REBS-based simulations perform so well. A smooth curve fit through random data would not result in large correlations only in no bias. The main reason for the good correlation here is that there is a trend in the time series, which is considerably large compared to the pollution events. REBS captures this trend very well. In contrast, GDB may contain a fair fraction of noise (as mentioned elsewhere) and even if the trend is correctly captured, this will lead to lower correlation.

We agree with this statement and made the following changes:

Regarding correlation, it is not surprising, that Table 2 shows the largest r² values for the REBS method, where the baseline is basically a fit of the observation data. → The RÉBS method shows the highest r² values. The main reason for this good correlation is that the method captures the trend in the time series very well, which represents a considerable fraction of the total variability in the data. The GDB baseline may contain a fair fraction of noise, in contrast to the smooth baselines of the other two methods. This will lead to lower correlation.

L373: But overall, the 10 d REBS has best correlation and MSE and bias are only slightly worse than for GDB at 50 days. It's a bit difficult to see the large benefits of the GDB method just from the statistics presented in Table 2.

Yes, we think that we get a better picture by viewing the statistics in Table 2 together with the emission increments (Figure 8). See also the rewritten text (comment to L385)

Figure 9: Consider using different y-scale for each country. Adding uncertainty estimates would be valuable as well.

We added the uncertainty estimates, however prefer to keep one y-scale for all countries (e.g. illustrating the dominant Chinese contribution). Furthermore, we show the absolute numbers above the bars, making the figure easy to understand.

[Figure]

Caption Fig 9, we added: Uncertainties represent a 1σ range.

L385: How can we conclude that the increments estimated with the REBS baseline are wrong? Based on the assumption that GDB is correct? Maybe simply formulate in a more careful way as done a few lines below for the Stohl baseline case.

Yes, we agree:

When using the REBS method (Fig. 8b), the inversion produces negative emission increments in almost all areas of the globe, indicating that calculated baselines are too high overall. This is consistent with the assumption that the method overestimates the baseline at individual stations by wrongly classifying observations as baseline observations that are actually influenced by emissions within the backward calculation period. In contrast, the inversion algorithm produces positive increments almost everywhere around the globe when applying Stohl's method (Fig. 8c), suggesting that the method systematically underestimates the baseline (not only at background stations) which generally leads to a priori emissions that are too high. In case of the GDB method (Fig. 8d) negative and positive increments are more balanced, showing no sign of a systematical under- or overestimation of the baseline. Large positive increments can be seen in East Asian regions and parts of Europe, whereas the inversion tends to produce slightly negative increments in the Southern Hemisphere. →

When using the REBS method (Fig. 8b), the inversion produces negative emission increments in almost all areas of the globe. As the real emissions are unknown, this is not necessarily an unrealistic result. However, when considering these mostly negative increments together with the discussed positive bias for REBS baselines in Table 2 (especially for longer backward simulation periods), there is reason to assume that the REBS method overestimates baselines and consequently underestimates the *a posteriori* emissions overall. In contrast, the inversion algorithm produces positive increments almost everywhere around the globe when applying Stohl's method (Fig. 8c). Again, considering this together with the discussed negative biases in Tab. 2, this might indicate an underestimation of the baselines and an overestimation of the *a posteriori* emissions overall. In case of the GDB method (Fig. 8d) negative and positive increments are more balanced. Overall the patterns are more similar to the ones of the REBS method, except in East Asia, where they rather resemble the patterns of Stohl's method. Large positive increments can be seen in East Asian regions and parts of Europe, whereas the inversion tends to produce slightly negative increments in the Southern Hemisphere.

L390: Overall I would mention that patterns in 8b and 8d are more similar than between 8c and 8d, with the exception of East Asia.

Changed accordingly (see comment to L385)

Figure 10: Consider adding posterior uncertainties.

 Changed accordingly:

[Figure]

Caption Fig 10, we added:  Uncertainties represent a 1σ range.

Figure 13: Consider adding information on backward integration time to the figure caption.

Added: for the 50-day simulation

L336ff: The conclusion that longer integration times allow for correct emission estimation in 'under-sampled' regions should be drawn more carefully. Yes ,technically this is true. However, the problem remains that the sensitivity to these regions (given the current network) is small compared to regions where the observations are taken. Small sensitivities may simply lead to more random adjustments in the inversion, because they would only contribute small changes to the observation mismatch. An observing system simulation experiment would be better suited to prove this point than analyzing uncertainty reductions and assuming that the 50 d inversion does the job correctly. Furthermore, one may get the idea that with the 50 d (or longer) integration we do not need any more flask sampling sites, something that is afterwards promoted in the manuscript.

We agree, that the discussion around the backward simulation period needs to be handled with more care. We have no doubt, that emission patterns in certain regions with small sensitivities cannot be determined accurately with reasonable spatial resolution and that a lack of observations cannot simply be compensated through "long enough" simulation periods. As this point was also brought up by reviewer #1 we took great care to make this clear in the manuscript.

We added at L488.

Final Remark

In this study, we show many advantages for using relatively long backward simulation periods for the inversion. Nevertheless, the improvement of regional emission patterns is still limited by the observation network. A lack of observations in one region cannot simply be compensated by extending the simulations for stations in other regions to very long periods. For backward simulation times of 20-50 days, the emission sensitivity is distributed over large areas but usually still concentrated within broad latitude bands. The additional information to be gained from such long simulation times, on top of the information provided by the shorter simulation times, can probably best be compared with the inversions done with a multi-box model such as the AGAGE 8-box model (e.g. Rigby et al. ;2013) that is capable of determining the emissions in broad latitude bands. Consequently, if the emissions in certain regions with a dense observation network are already well constrained by shorter simulation periods, the residual emission will be attributed correctly as an emission total to all other regions of the same latitude band with a poor station coverage. The effective resolution of the obtained emissions in such data-poor regions may be very coarse but the result might still be informative. Furthermore, the

emission sensitivity for the 20-50 day backward period is still not uniformly distributed over a latitude band and thus provides some limited regional information. Perhaps supported with a limited number of strategically located flask measurements, inversions with long backward simulation times could provide coarse but robust information on emissions in poorly sampled regions. Independently, the growing correlation between modeled and observed mixing ratios with increasing backward simulation length (Table 2; averaged over all stations) also shows that longer backward simulations hold additional information, even though the information gain decreases with every day added to the simulation length and probably becomes marginal for very long backward simulation times. However, we propose to make use of this additional information and apply longer periods whenever possible to make the best use of the existing observation network.

L447: The baseline could also be taken from nearby or same latitude continuous sites or, as in the method by Manning et al. (2021), could be represented through baselines at the domain border (not possible for global runs).

We added:

Here, the baseline could be taken from nearby or same latitude continuous sites, or represented through baselines at the domain border in case of regional inversions (Manning et al., 2021).

L450f: What is the impact of the additional flask data on the national and global emission estimates. Maybe add to Fig 10 and 11.

We rewrote this section and added additional text and figure to discuss the impact of the additional flask data on the national and global emission estimates.

[Figure]

Caption Fig 13: a) Relative change in *a posteriori* emissions and b) the additional error reduction when using flask measurements in addition to continuous measurements for the 50-day simulation. The locations of the flask measurements are marked with black dots.

Figure 13a) shows the relative change in a posteriori emissions and Fig. 13b) the additional relative error reduction when using flask measurements additionally to the continuous measurements for the 50-day backward simulation. One can see substantial differences in the USA, Eastern Europe, South Africa, East Asia and the Near East, where also an additional error reduction occurs. While this additional error reduction can be relatively large (up to 73 %) for grid cells in the vicinity of the measurement sites, it quickly decreases down to a few percent with larger distance to the measurements. Consequently, flask measurements show only small influence on the total global emission estimate (< 1%), but can have a large impact on calculated national emissions of specific countries (Fig. A4). For countries in the Near East the additional use of flask measurements changes national emission estimates by 40 to 100%. South African and American emissions are modified by around 10%.

[Figure]

Caption Fig A4: Relative change in national *a posteriori* emissions of selected countries, when flask measurements are used in addition to continuous measurements in the case of 50-day simulations.

L469ff and Fig 14c-f: I cannot follow the suggestion of how the bias in baseline is supposed to lead to the large bias in the global total emissions as given by the pink lines in Fig14. If I understand it correctly, the expectation is that the inflicted baseline bias would need to be compensated by increased emissions during the period of backward integration. Somehow, there seems to be some misconception here. Although, this consideration would make sense for a global (box) model that is run with baseline concentrations to estimate global emissions, it cannot be applied to the kind of observations and "regional" simulations done here. The sampled concentration peaks do not represent fully mixed emissions, but recent emission impacts. The bias of 0.003 ppt is orders of magnitude smaller than the regional emission signal simulated at the observation sites, even if only 1 day backward transport is considered (see Fig 4 and Fig 5). However, it is this regional signal that is used in the inversion step, not the annual global trend. Offsets in the regional signal in the order of 1 % will hardly have an effect on the emissions that is in the order suggested by the pink line in Fig 14. I would suggest to redo the sensitivity test with a biased background using a much larger bias then suggested here. How large the magnitude of this bias should be is hard to tell, but maybe it could be taken as the difference between REBS and Stohl baselines. Alternatively, a different interpretation and a re-thinking of the expectation (pink line) would also help these sensitivity tests.

The pink line is drawn simply to indicate an expected relationship between a baseline bias and a resulting emission bias if a global box model were used and the bias attributed solely to emissions in various periods. For very long backward simulation times, longer than 50 days, we expect our inversion to converge towards the pink line. For shorter simulation times, the reviewer is right that the inversion result depends heavily on the location of the stations and their proximity to emissions. The point here is exactly that for short simulation times, the inversion behavior is difficult to predict, while for long simulation times, the inversion attributes the bias to emissions that approach the value needed to explain the baseline bias. For 50 days backward simulation, we are already closely approaching this value. We have also tested much larger biases (see Fig A5) and here as well the inversion

shows this expected behavior that is consistent with the conservation of mass for long simulation times, while the result for short simulation times is unpredictable. As can be seen Fig. 14c/d and Fig. A5, for short simulation times the inversion sometimes even predicts emission changes that are of opposite sign from that expected by the added baseline bias, which we find unsatisfying. We argue that for long backward simulation times the inversion produces more robust results.

L463: Note here that for all these sensitivity cases we → Note that for all these sensitivity cases shown in Fig. 14 we

L475: To fully compensate the baseline bias equivalent to 3 days of emissions, global a posteriori emissions (dashed, pink line) would need to deviate strongly from the box model value for the 1 days case, but converge towards it with increasing backward simulation time. → To fully compensate the baseline bias equivalent to 3 days of emissions, global a posteriori emissions would need to deviate strongly from the reference value for the 1-days case, but converge towards it with increasing backward simulation time. This is shown by the dashed pink line, which indicates the expected relationship between this baseline bias and a resulting emission bias if a global box model was used and the bias attributed solely to emissions in different periods corresponding to the backward simulation times.

Caption Figure 14, we changed: The dashed red lines indicate the emissions that would result from attributing the global field bias in the global mixing ratio fields to emissions during the backward simulation period. → The dashed pink lines represent the expected relationship between the baseline bias and a resulting emission bias if a global box model was used and the bias attributed solely to emissions in different periods corresponding to the backward simulation times.

L485, we added: We also investigated the inversion behavior for larger baseline biases, subtracting/adding (Fig A5a/b) 0.05 ppt from/to the global fields, corresponding to roughly 50 days of the 2012 global $SF_6$ emissions. Here again, the results for short simulation times seem unpredictable, i.e. they do not follow the described expected behavior, indicated by the pink dashed lines. Only for the 50-days simulation periods results converge to the expected global emissions consistent with the respective baseline bias.

[Figure]

Figure A5. Global $SF_6$ emissions using the GDB method shown for two sensitivity tests, where a uniform bias of c) -0.05 ppt and d) +0.05 ppt is added to every grid cell of the global mixing ratio fields. Results are shown for backward simulation periods between 1 and 50 days, and for a 50-days backward simulation case, where additionally to continuous measurements also flask measurements were included in the inversion. The dashed pink lines represent the expected relationship between the baseline bias and a resulting emission bias if a global box model was used and the bias attributed solely to emissions in different periods corresponding to the backward simulation times.. For these two sensitivity tests, a priori uncertainties were set to 500%.

L492ff: Consider 'may lead' instead of 'leads to'. REBS baselines can still work as you show in Figure 10. It would be fair to mention here that the performance for integration periods that are typically used 10 or 20 days are in better agreement. Also the fact that most inversion systems try to optimize baseline biases as part of the inversion, should be mentioned.

leads to → may lead to

For instance, for the year 2012 inversions with the REBS method produce a posteriori global total $SF_6$ emissions of 9.8 Gg/yr and 3.2 Gg/yr for backward simulation periods of 1 day and 50 days, respectively, compared to a well known reference value of around 8.0 Gg/yr. → For instance, for the year 2012 inversions with the REBS method produce a posteriori global total $SF_6$ emissions ranging between 9.8 Gg/yr and 3.2 Gg/yr for backward simulation periods between 1 day and 50 days, compared to a well known reference value of around 8.0 Gg/yr. Optimizing the baseline shows little effect for simulation periods between 1 and 20 days, but could half the bias in the 50 day-simulation case. Although

the improvements of the baseline optimization increase with growing backward simulation period, the simultaneously growing bias cannot be compensated.

L497ff: Same as above: mention that baseline biases could be treated as part of the inversion.

We added: Optimizing the baseline, however, shows great improvements, especially for longer simulation periods.

L505ff: From Table 2 I can only conclude this for the GDB method. The other two methods show insignificant improvements from 10 to 50 days or even worse performance (in terms of bias). I would also think that this improved performance from 10 to 50 days for GDB would strongly depend on the baseline model. Here, a relatively coarse model is used. Higher resolution may result in very good performance already at shorter backward integration times.

From our point of view, improvements from 10 to 50 days can also clearly be seen with Stohl's method, but we agree that the statement might be too general. We also agree that a higher-resolution GDB model could lead to better correlations for shorter simulation periods.

Statistical comparisons of a priori modeled versus observed mixing ratios show that longer LPDM backward simulations outperform shorter simulations. In particular, extending the trajectory length from the usual 5-10 days to 50 days reduces the mean squared error and increases the correlation. → Statistical comparisons of a priori modeled versus observed mixing ratios suggest that longer LPDM backward simulations outperform shorter simulations. In particular, extending the trajectory length from 5-10 days to 50 days can reduce the mean squared error and increase the correlation.

L508: Again, I find this statement too general. The problem with the biased prior results from the fact that with short integration times there is no sensitivity to large areas, so there is no chance for the inversion to correct this. If observations would cover all emitting areas well within 10 days the bias should also be removed. Many regional and global scale inversions result exist where posterior emissions moved far away from the prior, but the key is observational constraint. If there is little constraint on certain elements of the state vector, we cannot expect the posterior result to be more accurate than the prior.

We agree that the key is observational constraint. However, we argue that the observational constraint on the global emissions is actually very strong even with only a few measurement sites. That's why annual mean global mean emissions can be determined quite accurately even with measurements from a single site and using a global box model. However, in our setup this strong constraint can only be utilized with long backward simulation times, whereas for short backward simulation times the global constraint does not come into play.

Inverse modelling is highly sensitive to biases in the a priori emissions as well as biases in the baseline. We could show that this sensitivity decreases with the length of the backward simulation period. While it is nearly impossible to correct biased global a priori emissions with backward simulation periods of 1-10 days, 50-day backward simulations can capture global emissions quite accurately even in the presence of large biases. → Inverse modeling is highly sensitive to biases in the a priori emissions as well as biases in

the baseline. We could show that this sensitivity can decrease with the length of the backward simulation period and we find that longer backward simulation periods can help to correct biased global emission fields. In the presented case, it is not possible to correct strongly biased global a priori emissions with backward simulation periods of 1-10 days, while they are captured quite accurately with 50-day backward simulations.

L515ff: This is very much in line with what Weiss et al. (2021) suggested as well. Why not mention that?

We added

Similar to Weiss et al. (2021) we suggest ….

L517ff: Both statements are very general. For regional inversions longer integration times don't necessarily make more sense. Baselines can be sampled from conditions at domain border, either from global model as in GDB or constructed from observations (like Manning et al., 2021). Similarly, the optimization of the baseline as part of the inversion to avoid biases, should be mentioned. This will still be necessary for the GDB approach when biased global fields are used.

Following these results, we strongly recommend to abandon the use of baseline methods based purely on the observations of individual sites, for inverse modeling. We also recommend to employ longer LPDM backward simulation periods, beyond the usual 5-10 days, as this leads to improvements in overall model performance, allows to constrain emissions in regions poorly covered by the monitoring network, and produces more robust global emission estimates. → Following these results, we advise against the use of baseline methods that are purely based on the observations of individual sites. At least great care needs to be taken that problems such as demonstrated in this paper do not occur. In order to reduce biases, the optimization of the baseline as part of the inversion might be necessary, but would likely not be sufficient to avoid biases completely. We recommend also to employ longer LPDM backward simulation periods, beyond 5-10 days, as this can lead to improvements in overall model performance, can produce more robust global emission estimates and might help to constrain emissions, at least at very coarse resolution, in regions poorly covered by the monitoring network.

**Technical corrections**

Equation 4: The last term should contain lower case x for the prior state vector.

➤ changed accordingly

**Additional references**

Colton, D., and Kress, R.: Inverse acoustic and electromagnetic scattering theory, Applied Mathematical Sciences, 93, 305, Berlin: Springer, https://doi.org/10.1007/978-3-662-02835-3 ,1998.

Groot Zwaaftink, C. D., Henne, S., Thompson, R. L., Dlugokencky, E. J., Machida, T., Paris, J.-D., Sasakawa, M., Segers, A., Sweeney, C., and Stohl, A.: Three-dimensional methane distribution simulated with FLEXPART 8-CTM-1.1 constrained with observation

data, Geoscientific Model Development, 11, 4469–4487, https://doi.org/10.5194/gmd-11-4469-2018, 2018.

Guillevic, M., Vollmer, M. K., Wyss, S. A., Leuenberger, D., Ackermann, A., Pascale, C., Niederhauser, B., and Reimann, S.: Dynamic gravimetric preparation of metrologically traceable primary calibration standards for halogenated greenhouse gases, Atmospheric Measurement Techniques, 11, 3351–3372, https://doi.org/10.5194/amt-11-3351-2018, 2018.

Koyama, Y., Maksyutov, S., Mukai, H., Thoning, K., and Tans, P.: Simulation of variability in atmospheric carbon dioxide using a global coupled Eulerian–Lagrangian transport model, Geoscientific Model Development, 4, 317–324, https://doi.org/10.5194/gmd-4-317-2011, 2011.

Leip, A., Skiba, U., Vermeulen, A., and Thompson, R. L.: A complete rethink is needed on how greenhouse gas emissions are quantified for national reporting, Atmos. Environ., 174, 237-240, doi: 10.1016/j.atmosenv.2017.12.006, 2018.

Manning, A. J., Redington, A. L., Say, D., O'Doherty, S., Young, D., Simmonds, P. G., Vollmer, M. K., Mühle, J., Arduini, J., Spain, G., Wisher, A., Maione, M., Schuck, T. J., Stanley, K., Reimann, S., Engel, A., Krummel, P. B., Fraser, P. J., Harth, C. M., Salameh, P. K., Weiss, R. F., Gluckman, R., Brown, P. N., Watterson, J. D., and Arnold, T.: Evidence of a recent decline in UK emissions of hydrofluorocarbons determined by the InTEM inverse model and atmospheric measurements, Atmos. Chem. Phys., 21, 12739-12755, doi: 10.5194/acp-21-12739-2021, 2021.

Nisbet, E., and Weiss, R.: Top-Down Versus Bottom-Up, Science, 328, 1241, doi: 10.1126/science.1189936, 2010.

O'Doherty, S., Simmonds, P. G., Cunnold, D. M., Wang, H. J., Sturrock, G. A., Fraser, P. J., Ryall, D., Derwent, R. G., Weiss, R. F., Salameh, P., Miller, B. R., and Prinn, R. G.: In situ chloroform measurements at Advanced Global Atmospheric Gases Experiment atmospheric research stations from 1994 to 1998, J. Geophys. Res., 106, 20429-20444, doi: 10.1029/2000JD900792, 2001.

Weiss, R. F., and Prinn, R. G.: Quantifying greenhouse-gas emissions from atmospheric measurements: a critical reality check for climate legislation, Philosophical Transactions of the Royal Society A: Mathematical, Physical and Engineering Sciences, 369, 1925-1942, doi: 10.1098/rsta.2011.0006, 2011.

---

## Author Response (AR2)

**Author's Response**

**Comments to the author**:

Dear Authors,

I am pleased to inform you that the revised version of your manuscript is accepted for publication after some technical correction. Your response to the referees' comments and your efforts to address the raised criticism are very much appreciated.

As you can see from the reviewer comment below, there is still some disagreement regarding the impact of the simulation length, but as mentioned in the comment it is up to you whether you want to soften your statements in this respect or not.

We would like to thank Andrea Stenke a lot for editing our manuscript!

In the response we use 4 different colors. The blue colored text is the general answer to the reviewer's comments. Additionally, we show how the text is changed in the manuscript: The original text is colored grey, removed text is colored red, and new text is colored green.

We softened our statement:

L512)

Comparing the inversion results for doubled (Fig. 14a) and halved (Fig. 14b) a priori emissions clearly shows that the corresponding biases in the global a posteriori emissions are reduced substantially with increasing backward simulation period and converge towards the rather well known global SF6 emission from the box model. However, it is clear that a substantial bias remains even with a backward simulation period of 50 days. It seems likely that an extension of the backward simulation period beyond 50 days would further reduce the bias. - >

It seems a further extension of the backward simulation period beyond 50 days would be required in order to reduce the remaining bias.

Technical correction: In line 549 it says "the AGAGE 8-box model". This should be changed to 12-box model. Please correct also other instances, which I might have overlooked.

L549)

AGAGE 8-box model -> AGAGE 12-box model